



# Representing the impact of *Rhizophora* mangroves on flow and sediment transport in a hydrodynamic model (COAWST_rh v1.0): the importance of three-dimensional root system structures

Masaya Yoshikai[1], Takashi Nakamura[1], Eugene C. Herrera[2], Rempei Suwa[3], Rene Rollon[4], Raghab Ray[5], Keita Furukawa[6], and Kazuo Nadaoka[1]

[1]School of Environment and Society, Tokyo Institute of Technology, Tokyo 152-8552, Japan
[2]Institute of Civil Engineering, University of the Philippines, Diliman, Quezon City 1101, Philippines
[3]Forestry Division, Japan International Research Center for Agricultural Sciences (JIRCAS), Ibaraki 305-8686, Japan
[4]Institute of Environmental Science & Meteorology, College of Science, University of the Philippines, Diliman, Quezon City, 1001, Philippines
[5]Atmosphere and Ocean Research Institute, The University of Tokyo, Chiba, 277-8564, Japan
[6]NPO Association for Shore Environment Creation, Kanagawa, 220-0023, Japan

*Correspondence to*: Masaya Yoshikai (yoshikai.m.aa@m.titech.ac.jp)

**Abstract.** In hydrodynamic models, vegetation is commonly approximated as an array of vertical cylinders to represent its impacts on flow and sediment transport. However, this simple approximation may not be valid in the case of *Rhizophora* mangroves that have complicated three-dimensional root structures. Here, we present a new model to represent the impacts of *Rhizophora* mangroves on flow and sediment transport in hydrodynamic models. The model explicitly accounts for the effects of the three-dimensional root structures on flow and turbulence, as well as the effects of two different length scales of vegetation-generated turbulence characterized by stem diameter and root diameter. The model employs an empirical model for the *Rhizophora* root structures that can be applied using basic vegetation parameters (mean stem diameter and tree density), without rigorous measurements of the root structures. We showed that compared to the conventional approximation using an array of cylinders, the new model significantly improves the predictability of velocity, turbulent kinetic energy, and bed shear stress measured in a model and a real *Rhizophora* mangrove forest. The model further suggested the high efficiency of the three-dimensional root structures of *Rhizophora* mangroves on sedimentation, which allows a relatively high sediment supply to the forest but effectively regulates sediment erosion through reduced bed shear stress, compared to cylinder arrays that exhibit equivalent sediment supply or sediment retention. The presented model could be a fundamental tool to advance our understanding of the sedimentary processes in *Rhizophora* mangrove forests which are linked to mangroves' vulnerability and ecosystem service.

## 1 Introduction

Mangroves are one of the coastal wetland ecosystems that grow in intertidal areas in tropical and subtropical regions (Hamilton and Casey, 2016). They have characteristic aboveground root systems of which morphological structures





considerably vary among genera, such as pneumatophores or "pencil roots" of *Avicennia* and *Sonneratia* genera, and prop
roots of *Rhizophora* genus (Krauss et al., 2014). Especially due to the presence of aboveground root systems, mangroves exert
drag against water flow that slows down flow velocity and creates conditions preferable for the deposition and retention of
tidally and fluvially transported sediments (Furukawa et al., 1997; Krauss et al., 2003; Horstman et al., 2015; Chen et al., 2016,
2018; Willemsen et al., 2016; Best et al., 2022) similar to other wetland vegetation habitats such as salt marshes (Temmerman
et al., 2005; Bouma et al., 2007; Mudd et al., 2010; Weisscher et al., 2022). The vegetation-flow-sediment interaction is
considered a major driving factor of the long-term geomorphic evolution of wetland vegetation habitats (Mariotti and
Fagherazzi, 2010; Mariotti and Canestrelli, 2017; Brückner et al., 2019; Kalra et al., 2022; Willemsen et al., 2022) that enabled
them to counteract the threats due to sea-level rise (Fagherazzi et al., 2012, 2020; Lovelock et al., 2015; Kirwan et al., 2016).

Aside from counteracting the effects of sea-level rise, mangroves' drag effects and sediment retention function are closely
related to the major ecosystem services that they provide. Mangrove forests are known as globally significant carbon sinks due
to the high efficiency of burial and preservation of organic carbon in their sediments (Donato et al., 2011; Mcleod et al., 2011;
Sharma et al., 2020). Depending on the geophysical settings, organic carbon transported from outer systems could account for
a significant fraction of carbon stored in mangrove sediments (Sasmito et al., 2020; Suello et al., 2022) suggesting the vital
importance of sediment transport and deposition for carbon sequestration in mangrove forests. Given their great relevance to
mangroves' ecosystem services and vulnerability, a better understanding of the sedimentary processes in mangrove forests
such as transport, deposition, and retention as well as long-term geomorphic evolution is urgently needed to effectively
conserve and restore mangrove forests, which have degraded globally due to deforestation (Friess et al., 2019).

Hydrodynamic models that simulate flow and sediment transport processes including deposition and retention could be a
key tool to evaluate these sedimentary processes (e.g., Lokhorst et al., 2018), where the representation of vegetation drag is
essentially important (Temmerman et al., 2005; Nardin et al., 2016). Several modeling studies have shown that depending on
the magnitude of vegetation drag, the resulting geomorphic evolution can vary dramatically, hence the ecosystems' fate in
response to sea-level rise (Boechat Albernaz et al., 2020; Xie et al., 2020). The vegetation drag in salt marshes and seagrass
beds is commonly represented in hydrodynamic models by an array of vertical cylinders (cylinder drag model; Ashall et al.,
2016; Zhu et al., 2020), the drag effect of which has been well documented in both for emergent and submerged cases by a
number of studies (e.g., Nepf, 1999, 2012). Although fewer compared to studies on salt marshes, some modeling studies have
evaluated flow and sediment transport processes in mangrove forests (van Maanen et al., 2015; Bryan et al., 2017; Rodríguez
et al., 2017; Xie et al., 2020). However, most of them are limited to *Avicennia* or *Sonneratia*-dominated mangrove forests
whose aboveground roots (pencil roots) are geometrically simple and resemble that of a cylinder array.

In contrast, the root system of *Rhizophora* genus (prop root system) has three-dimensionally complicated structures that
may not be simply approximated by the array of vertical cylinders. Consequently, the representation of drag by *Rhizophora*
mangroves in hydrodynamic models remains to be established despite the worldwide occurrences of this mangrove genus
(Friess et al., 2019). This knowledge gap can be seen in the studies that have approximated the drag by *Rhizophora* mangroves



with arbitrarily increased bed roughness (Zhang et al., 2012) or cylinder arrays with arbitrary cylinder density (Xie et al. 2020) without much theoretical and experimental support. One exception is a modeling study by Horstman et al. (2015) that approximated the root structures of *Rhizophora* mangroves using a cylinder array with vertically variable cylinder densities. However, their method requires an exhausting field survey of the root structures as a requirement for proper model application, which may not be feasible for a forest-scale simulation.

In addition to flow velocity, vegetation affects turbulence (Nepf, 2012; Xu and Nepf, 2020), which is also relevant to the transport of substances (e.g., sediment and solutes) through turbulent diffusion (Tanino and Nepf, 2008; Xu and Nepf, 2021). While several hydrodynamic models can account for the vegetation-generated turbulence for the turbulence closure (e.g., Temmerman et al., 2005, Marsooli et al., 2016), so far, no model has been established to predict the turbulence structures in *Rhizophora* mangrove forests. A rigorous, but feasible representation of the impact of *Rhizophora* mangroves on flow velocity
and turbulence in a hydrodynamic model is thus needed for a better understanding of sedimentary processes in *Rhizophora* mangrove forests.

Recently, insights on flow in *Rhizophora* mangrove forests have been increasing from laboratory- and field-based studies (e.g., Zhang et al., 2015; Maza et al., 2017; Shan et al., 2019; Yoshikai et al., 2022a). The availability of these studies allows us to formulate and test the representation of impacts of *Rhizophora* mangroves on flow in hydrodynamic models. Insights on
root structures have also been increasing from studies that have measured root structures in relation to tree size (Ohira et al., 2013; Yoshikai et al., 2021; Mori et al., 2022). These studies have shown the possibility of predicting root structures from limited vegetation parameters such as stem diameter, which would make the model feasible to apply in a forest-scale simulation with complicated root structures that are challenging to quantify.

In order to contribute to accurate but feasible simulations of flow and sediment transport in *Rhizophora* mangrove forests,
here, we implement a new model to represent the impacts of *Rhizophora* mangroves in a three-dimensional hydrodynamic model–the Regional Ocean Modeling System (ROMS; Shchepetkin and McWilliams, 2005) of the model framework COAWST (Coupled Ocean–Atmosphere–Wave–Sediment Transport Modeling System; Warner et al., 2010). The impact of vertically varying projected area of roots on flow velocity and turbulence is specifically taken into consideration by the new model. Furthermore, the new model accounts for two different length scales of turbulence generated by *Rhizophora*
mangroves–stem diameter and root diameter–as characterized by a flume experiment by Maza et al. (2017). We also incorporate an empirical *Rhizophora* root model proposed by Yoshikai et al. (2021) into the hydrodynamic model for a practical application to *Rhizophora* mangrove forests for which root structures are unknown. Here, we aim to examine the following: (a) how does the new representation of *Rhizophora* mangroves in the hydrodynamic model improve the predictability of flow velocity and turbulence compared to the conventional drag approximation using cylinder arrays? (b) how can the new model
be effectively applied for an accurate prediction of the flow in *Rhizophora* mangrove forests by incorporation of the *Rhizophora* root model? After addressing these questions, we examined (c) how the *Rhizophora* mangroves could affect sediment transport, deposition, and retention using a numerical experiment on sediment transport.





## 2 Materials and Methods

### 2.1 Model description

A proposed framework for modeling the flow and sediment transport in *Rhizophora* mangrove forests is presented in Fig. 1. We used ROMS and Community Sediment Transport Model (CSTM; Warner et al., 2008) under the model framework COAWST. The vegetation module has been added by Beudin et al. (2017) to account for the drag by vegetation, such as seagrasses and salt marshes, in the momentum equations in ROMS, where the formulations added are basically in the same form as the cylinder drag model (see Text S1). We modified the equations introduced by Beudin et al. (2017) to make them

suitable for representing the impact of *Rhizophora* mangroves on flow and thereby sediment transport; these equations are described below. We added a new module in COAWST–*Rhizophora* root module–that provides the vertical profile of projected area density of root systems from stem diameter and tree density in each model grid (Fig. 1).

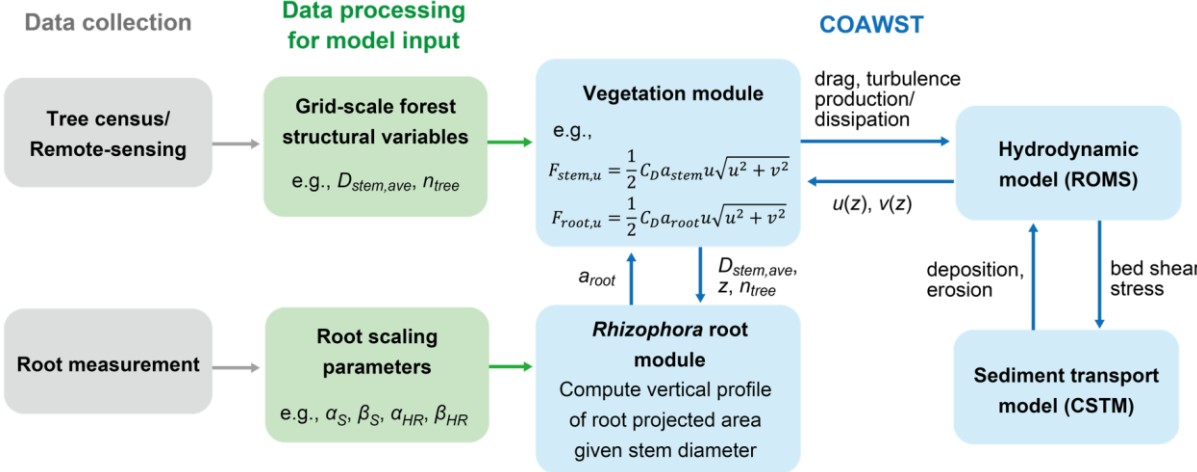

Figure 1. The proposed framework of modeling flow and sediment transport in *Rhizophora* mangrove forests using COAWST.

$D_{stem,ave}$ and $n_{tree}$ are the mean stem diameter and tree density, respectively, to be given in each grid, $a_{stem}$ and $a_{root}$ are the stem and root projected area density, where $a_{stem}$ is a product of $D_{stem,ave}$ and $n_{tree}$. $F_{stem,u}$ and $F_{root,u}$ are the drag forces exerted on $u$-component of flow by the stem and root, respectively. See Text S3 and Table S1 for explanations of the root scaling parameters.

    This manuscript basically considers velocities as temporally averaged ones unless specified. We did not consider the subgrid-scale spatial heterogeneity of velocity generated by vegetation as in other modeling studies (e.g., King et al., 2012;

Marsooli et al., 2016). The Reynolds number (*Re*) defined using the root diameter as length-scale suggested fully turbulent structures of root-generated wakes (*Re* > 120; Shan et al., 2019) even for weak currents (~1 cm s⁻¹) that could diminish the dependence of drag coefficient ($C_D$) on *Re*. Thus, we treat $C_D$ as a constant as in Beudin et al. (2017). For simplicity, we present



equations in the two-dimensional form on *x-z* plane (zero velocity in *y*-direction) while the equations implemented in ROMS are three-dimensional (*x-y-z*), where *x-y* represents the horizontal plane and *z* represents the vertical direction.

**2.1.1 Drag force**

In *Rhizophora* mangrove forests, the stem and roots are the main components that exert drag in tidal flows. We partition the drag by *Rhizophora* mangroves (vegetation drag) into the contributions by stems and roots and calculated it using the quadratic drag law as

$$F_{veg}(z) = F_{stem}(z) + F_{root}(z) = \frac{1}{2}C_D n_{tree} D_{stem,ave} u(z)^2 + \frac{1}{2}C_D a_{root}(z)u(z)^2 \tag{1}$$

where $F_{veg}$ is the spatially-averaged vegetation drag (m s$^{-2}$), $z$ is the height from bed (m), $F_{stem}$ and $F_{root}$ are the contributions by stems and roots to $F_{veg}$, respectively, $C_D$ is the drag coefficient, $n_{tree}$ is the tree density (m$^{-2}$), $D_{stem,ave}$ is the mean stem diameter (m), $a_{root}$ is the spatially-averaged projected area density of roots (m$^{-1}$), and $u$ is the flow velocity (m s$^{-1}$). We represented stems as cylindrical shapes with vertically uniform diameter (Maza et al., 2017) then calculated the $F_{stem}$ using the cylinder drag model–the same equations introduced by Beudin et al. (2017) (Text S1 and Table 1). Here, we assumed the

vertically constant and uniform drag coefficient ($C_D$) for stems and roots.

**2.1.2 Turbulence**

In ROMS, the generic length-scale (GLS) model is implemented as the turbulence closure, where the equations can represent several two-equation closure models such as *k–ε* and *k–ω* models by adjusting the model parameters (Umlauf and Burchard, 2003; Warner et al., 2005). In this manuscript, we present equations in the form of *k–ε* model for reference purposes

as this is the most studied two-equation closure model for flows in vegetated areas (López and García, 2001; Katul et al., 2004; Defina and Bixio, 2005; King et al., 2012). Beudin et al. (2017) have included an additional term for wake production due to vegetation ($P_w$) in the equation for turbulence kinetic energy (TKE) as

$$\frac{\partial k}{\partial t} + u\frac{\partial k}{\partial x} = \frac{\partial}{\partial z}\left(\frac{v_t}{\sigma_k}\frac{\partial k}{\partial z}\right) + P_s + B + P_w - \varepsilon \tag{2}$$

where $k$ is TKE (m$^2$ s$^{-2}$), $v_t$ is the eddy viscosity (m$^2$ s$^{-1}$), $\sigma_k$ is the turbulent Schmidt number for $k$ (1.0), $P_s$, $B$, and $P_w$ represent

the production of $k$ by shear, buoyancy, and wakes generated by vegetation (m$^2$ s$^{-3}$), respectively, and $\varepsilon$ is the turbulent dissipation (m$^2$ s$^{-3}$). Similarly, they included an additional term ($D_w$) in the equation for $\varepsilon$ as

$$\frac{\partial \varepsilon}{\partial t} + u\frac{\partial \varepsilon}{\partial x} = \frac{\partial}{\partial z}\left(\frac{v_t}{\sigma_\varepsilon}\frac{\partial k}{\partial z}\right) + \frac{\varepsilon}{k}(c_1 P_s + c_3 B - c_2 \varepsilon) + D_w \tag{3}$$

where $\sigma_\varepsilon$ is the turbulent Schmidt number for $\varepsilon$ (1.3), $c_1$ (1.44), $c_2$ (1.92), and $c_3$ are the model constants, where the value of $c_3$ varies depending on stratification state (Warner et al., 2005), and $D_w$ is the dissipation rate of wakes (m$^2$ s$^{-4}$). The wake

production rate ($P_w$) is typically considered equal to the rate of work done by the flow against vegetation drag, i.e., $P_w = F_{veg}u$



(Nepf, 2012). In contrast, the turbulence dissipation rate largely depends on the turbulence length-scale in addition to the TKE, which requires a prior knowledge of the turbulence length-scale of wakes for correctly predicting the $D_w$ (King et al., 2012; Liu et al., 2017; Li and Busari, 2019).

Previous flume studies for flow through vegetated area have shown that the stem diameter (or leaf width) is the plausible turbulence length-scale of wakes (Tanino and Nepf, 2008; King et al., 2012). In the case of flow in *Rhizophora* mangrove forests, however, there are two potential length-scales–stem diameter and root diameter, which could significantly differ from each other (Maza et al., 2017). This variation makes it challenging to parameterize them into one representative length-scale of wakes ($L$ in Eq. S6 in Text S2). To resolve this, we partitioned the $P_w$ and $D_w$ into the terms for wakes generated by stems and roots, respectively, as

$$P_w = P_{w,stem} + P_{w,root} = F_{stem}u + F_{root}u \tag{4}$$

$$D_w = D_{w,stem} + D_{w,root} = c_2 \frac{P_{w,stem}}{\tau_{stem}} + c_2 \frac{P_{w,root}}{\tau_{root}} \tag{5}$$

where $P_{w,stem}$ and $P_{w,root}$ (m$^2$ s$^{-3}$) are the production of $k$ by stem- and root-generated wakes, $D_{w,stem}$ and $D_{w,root}$ (m$^2$ s$^{-4}$) are the dissipation rate of stem- and root-generated wakes, and $\tau_{stem}$ and $\tau_{root}$ (s) are the time-scale of stem- and root-generated wakes, respectively; these are given by

$$\tau_{stem} = \left( \frac{L_{stem}^2}{c_w^2 P_{w,stem}} \right)^{1/3} \tag{6a}$$

$$\tau_{root} = \left( \frac{L_{root}^2}{c_w^2 P_{w,root}} \right)^{1/3} \tag{6b}$$

where $L_{stem}$ and $L_{root}$ (m) are the length-scale of stem- and root-generated wakes, respectively, and $c_w$ is the model constant. Here, we gave mean stem diameter ($D_{stem,ave}$) and root diameter ($D_{root,ave}$) to $L_{stem}$ and $L_{root}$, respectively.

We considered $c_w$ in Eq. (6) as a calibration parameter whereas Beudin et al. (2017) gave a value of 0.09. Tanino and Nepf (2008) predicted the TKE for a flow through array of emergent cylinders with cylinder projected area density, $a$, and cylinder diameter, $d$, using $k = \gamma \left( \frac{1}{2} C_D a d \right)^{2/3} u^2$, where $\gamma$ is the scale coefficient that needs to be empirically determined. We can relate $c_w$ with $\gamma$ as $c_w = \gamma^{-3/2}$ by applying the $k$–$\varepsilon$ model to a limiting case of a steady, uniform, and neutrally-stratified flow through homogeneous emergent vegetation such that all the terms in Eqs. (2–3) except for $k$, $\varepsilon$, $P_w$, and $D_w$ can be neglected (King et al., 2012; Liu et al., 2017). We adjusted the value of $c_w$ so that the corresponding $\gamma$ value falls within a reported range (0.8–1.6; King et al., 2012; Xu and Nepf, 2020).



### 2.1.3 Root projected area density

We used the empirical *Rhizophora* root model (*Rh*-root model) developed by Yoshikai et al. (2021) as a predictor of the root projected area density ($a_{root}$) in Eq. (1). Based on an allometric relationships characterized by some site- and species-specific root scaling parameters ($\alpha_S$, $\beta_S$, $\alpha_{HR}$, $\beta_{HR}$ in Eq. S7 in Text S3), the *Rh*-root model predicts the vertical profile of root

projected area per vertical interval ($dz$; 0.05 m in this study) for a tree "*i*" ($A_{root,i}(z)$ (m$^2$)), where the subscript "*i*" represents the tree index, from the stem diameter of the tree "*i*" ($D_{stem,i}$). In short, $A_{root,i}(z)$ is expressed as $A_{root,i}(z) = f(D_{stem,i})$, where $f$ represents a function of the *Rh*-root model (see Text S3 for the details).

The vertical profile of spatially-averaged projected area density of roots in each grid can be calculated as $a_{root}(z) = n_{tree} \sum_{i=1}^{N_{tree}} f(D_{stem,i})/(N_{tree} dz)$, where $n_{tree}$ is the tree density (m$^{-2}$) and $N_{tree}$ is the number of trees in each grid. While some

variations in tree sizes (i.e., $D_{stem,i}$) and thereby $f(D_{stem,i})$, within a grid are expected, it would be convenient if the subgrid-scale variations can be parameterized using a grid-scale parameter for modeling purpose. In this study, we propose that the mean stem diameter ($D_{stem,ave}$) can be used for the parameterization as $\sum_{i=1}^{N_{tree}} f(D_{stem,i})/N_{tree} \approx f(D_{stem,ave})$, so that $a_{root} \approx n_{tree} f(D_{stem,ave})/dz$.

We investigated the above assumption using tree census data collected from three sites (Bak1, Bak2, and Fuk; see Text 4

for the site description). Using the *Rh*-root model, we computed the vertical distribution of the mean projected area of individuals in the tree census plots, $\sum_{i=1}^{N_{tree}} f(D_{stem,i})/N_{tree}$, and its representation using the mean stem diameter, $f(D_{stem,ave})$, and compared them (Fig. 2). This demonstrates that the use of $D_{stem,ave}$ can well represent the mean projected area density of individuals for all the three sites regardless of the differences in the forest structures (e.g., stem diameter distribution, tree density) and root scaling parameters (Table S1).

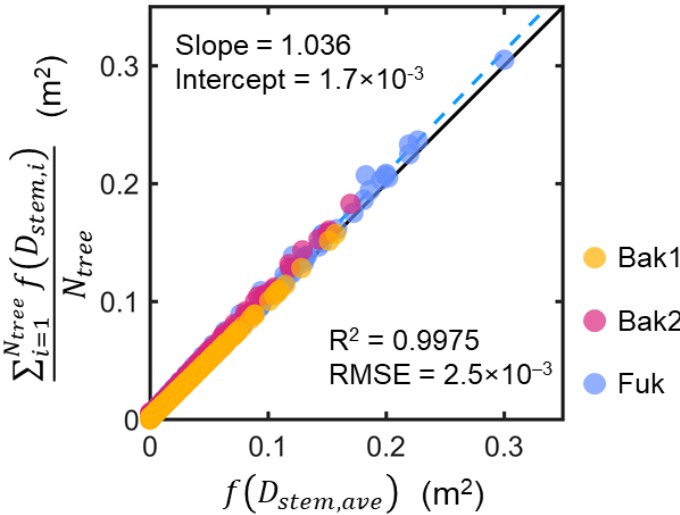




Figure 2. Comparison of the vertical profiles of mean projected area per vertical height interval ($dz$; 0.05 m) of individuals in tree census plots from three sites (Bak1, Bak2, and Fuk), $\sum_{i=1}^{N_{tree}} f(D_{stem,i})/N_{tree}$, and its representation using the mean stem diameter, $f(D_{stem,ave})$, where $N_{tree}$ is the number of *Rhizophora* trees in a plot, $D_{stem,ave}$ is the mean stem diameter of *Rhizophora* trees in the plot, the subscript "$i$" represents the tree index, and $f$ represents the function of the *Rhizophora* root model that gives the vertical profile of the root projected area of individuals.

### 2.1.4 Implementation to the COAWST

We replaced the equations for drag force and turbulence introduced by Beudin et al. (2017) with the ones presented above (Eqs. (1)–(6)) in the COAWST. The *Rhizophora* root module that gives $a_{root}(z)$ from $n_{tree}$ and $D_{stem,ave}$ using the *Rh*-root model was newly added in the COAWST (Fig. 1). Table 1 shows the grid-explicit input parameters related to this study. Parameters related to root structures are inputted to the model as universal parameters (not grid-explicit; Table S1). We introduced a new input parameter, species index ($spe$), that identifies the vegetation as *Rhizophora* species ($spe = 1$) or seagrass/marsh species ($spe = 0$). Depending on $spe$, the model interpretation of the inputted parameters varies (Table 1). If $spe = 1$, the vegetation module interacts with the *Rhizophora* root module for $a_{root}$ (Fig. 1) to compute the drag by the roots ($F_{root}$ in Eq. (1)) and the TKE production and dissipation of the root-generated wakes ($P_{w,root}$ and $D_{w,root}$ in Eqs. (4)–(6)). Otherwise ($spe = 0$), zero value is given to $a_{root}$, which vanishes all the root-related terms in Eqs. (1), (4)–(6), making them identical to the ones introduced by Beudin et al. (2017) and thus applying the cylinder drag model (however, see Text S2 for the modification of the equations of Beudin et al. (2017)). This means that the equations presented can be used both for *Rhizophora* mangroves and seagrasses/marshes by switching the value of $spe$.

Table 1. Grid-explicit input parameters. Symbols used in Beudin et al. (2017) are also shown. Parameters absent in the column of Beudin et al. (2017) are newly added in this study. Mean tree height ($H_{ave}$) is only relevant for some extreme conditions when the water level becomes higher than $H_{ave}$.

| Symbol | | Unit | Interpretation by the model | |
|---|---|---|---|---|
| This study | Beudin et al. (2017) | | Case $spe = 1$ | Case $spe = 0$ |
| $spe$ | | - | *Rhizophora* species | Seagrass/marsh |
| $D_{stem,ave}$ | $b_v$ | m | Mean stem diameter | Leaf width or stem diameter |
| $n_{ree}$ | $n_v$ | m$^{-2}$ | Tree density | Leaf or stem density |
| $H_{ave}$ | $l_v$ | m | Mean tree height | Leaf or stem length |





## 2.2 Model testing

We tested the model against the measurements of flows in a model *Rhizophora* mangrove forest by Maza et al. (2017) and in a real planted *Rhizophora* mangrove forest by Yoshikai et al. (2022a). Both studies have provided detailed information

of vegetation and hydrodynamic parameters that allow us to evaluate the model performance.

We created an orthogonal computational grid of 200 m × 200 m area with 5 m horizontal resolution for the model runs. We set the bed elevation and vegetation parameters uniform over the model domain both for the model- and real-mangrove forest test cases. To create a unidirectional flow in the model, we set the eastern and western boundaries of the model domain as closed (no water fluxes) and the northern and southern boundaries as open. Then we imposed water level differences

between the northern and southern boundaries to drive the flow based on the pressure gradient, where the water fluxes through the boundaries are given to equate the local pressure gradient and the drag force (bed + vegetation). The model was run with no wind conditions. When the steady state of flow was attained in the simulation, we compared the flow condition at the center of the model domain with the measurement. Table 2 summarizes the key vegetation and hydrodynamic parameters for each test case. Below we describe overview of the measurements by Maza et al. (2017) and Yoshikai et al. (2022a) and the model

settings.

Table 2. Vegetation and hydrodynamic parameter settings for model testing against flume experiments (Exp 1 and 2) in Maza et al. (2017) and field measurement in Yoshikai et al. (2022a). The row for $\gamma$ shows the values that best fit the measurements within the range of 0.8–1.6.

| Parameter | Exp 1 | Exp 2 | Field |
|---|---|---|---|
| Stem diameter ($D_{stem}$, m) | 0.2 | 0.2 | 0.066[a] |
| Root diameter ($D_{root}$, m) | 0.038 | 0.038 | 0.030[a] |
| Maximum root height ($HR_{max}$, m) | 2.01 | 2.01 | 1.10[a] |
| Tree density ($n_{tree}$, m$^{-2}$) | 0.072 | 0.072 | 0.36 |
| Drag coefficient ($C_D$) | 0.8 | 0.8 | 1.0 |
| Bottom roughness ($z_0$, m) | $0.5 \times 10^{-3}$ | $0.5 \times 10^{-3}$ | $0.5 \times 10^{-3}$ |
| Water depth ($h$, m) | 3.0 | 1.79 | 0.14–0.53[b] |





| | | | |
|---|---|---|---|
| Cross-sectional mean velocity ($U$, m s$^{-1}$) | 0.31 | 0.58 | [c] |
| Scale coefficient ($\gamma$) | 1.5 | 0.9 | 0.8 |

[a] Mean value at the measurement site.

[b] Water depth varies depending on tidal phase (see Fig. 6a, e).

[c] One of the target parameters for model prediction.

### 2.2.1 Application to a laboratory-based study

The model *Rhizophora* mangrove forest created in the flume by Maza et al. (2017) was 1/12$^{th}$, while we ran our model in a real-scale, e.g., we converted the velocities in the flume to the real-scale by keeping the Froude number (Table 2). The real-

scale vertical profile of vegetation projected area density ($a$) is shown in Fig. 3a. Maza et al. (2017) fabricated the root systems based on the data in Ohira et al. (2013) and distributed the model trees in-line in the flume. Maza et al. (2017) created two flow conditions by varying the water depth ($h$) and cross-sectional mean velocity ($U$) (Exp 1 and 2; Table 2) and measured the vertical profiles of velocities and TKE at five lateral positions in the model forest, at which flows were fully developed. We averaged the data taken at the five positions to estimate the spatial average of the velocity and TKE to be compared with the

model output.

We run the model with 15 vertical layers with approximately uniform layer thickness. We imposed the real-scale vertical profile of $a$ examined in Maza et al. (2017) (black markers in Fig. 3a) over the model domain. This means that the *Rhizophora* root module that predicts $a_{root}$ was not applied for the simulations performed here. We optimized the water levels at the boundaries to create the same flow conditions ($h$ and $U$) at the center of the model domain as the ones in Exp 1 and 2,

respectively.

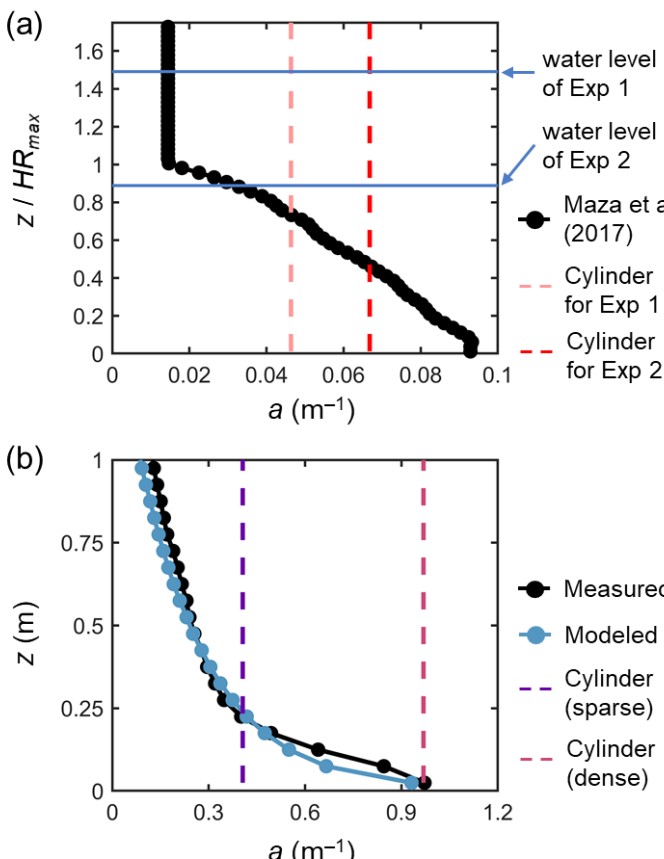

Figure 3. Vertical profiles of vegetation projected area density, *a*, in (a) a model *Rhizophora* mangrove forest examined by Maza et al. (2017) and (b) a real *Rhizophora* mangrove forest examined by Yoshikai et al. (2022a), where the values were calculated with $dz = 0.05$ m vertical interval (markers). $HR_{max}$ is the maximum root height (2.01 m in Maza et al. (2017); Table

1). The modeled *a* in panel "b" is given by the *Rhizophora* root module using the parameters shown in Tables 2 and S1 (for Bak2). The projected area density of cylinder arrays, which were used for comparison with the new model to represent the impacts of *Rhizophora* mangroves is also shown (dashed lines).

We used a value 0.8 for the drag coefficient ($C_D$) in the model (Table 2), the value of which was derived in fully developed flows with high Reynolds numbers (> 230) by Maza et al. (2017). The value of the bottom roughness ($z_0$) in the flume is

unknown; hence we gave 0.5 mm to $z_0$, which is the value derived in the field-based study (see Section 2.2.2). Due to the uncertainty of the bottom roughness, we did not include the modeled near-bed velocity and TKE for comparison with the measurements, above which the velocity and TKE were insensitive to the bottom roughness. Another unknown parameter is the scale coefficient in the turbulence closure, $\gamma$ (= $c_w^{2/3}$; see Section 2.1.2). We run the model by varying $\gamma$ in a reported range (0.8–1.6) with an interval 0.1 to seek a value that produced the best fit with the measurement, mostly for the TKE profile.



In addition to the simulation using the actual $a$ (black markers in Fig. 3a), referred to as "$Rh$ model", we tested the use of the cylinder drag model, referred to as "cylinder model". We defined the cylinder array for Exp 1 and 2, respectively, where the cylinder projected area density was set equal to the depth-average of the actual $a$ for each case (dashed lines in Fig. 3a); we set the cylinder diameter equal to the root diameter of the model *Rhizophora* trees (0.038 m; Table 1).

### 2.2.2 Application to a field-based study

We applied the model to a field mangrove setting examined by Yoshikai et al. (2022a) to investigate the effectiveness of the proposed framework (Fig. 1) for reproducing the observed flow conditions. Yoshikai et al. (2022a) measured vegetation and hydrodynamic parameters at 17-year-old planted stands of *Rhizophora apiculata* in a mangrove forest locally known as Bakhwan Ecopark in Aklan, Philippines. The site corresponds to Bak2 in Fig. 2. Like in the flume condition of Maza et al. (2017), approximately uniformly sized trees are evenly distributed. The measured spatially averaged vegetation projected area

density ($a$) at the site is shown in Fig. 3b. Due to the higher complexity of the root systems and higher tree density (Table 2), the $a$ near the bed showed almost 10 times higher value than the one in Maza et al. (2017) (Fig. 3). Yoshikai et al. (2022a) conducted hydrodynamic measurements during ebb tides on September 10 and 11, 2018 that corresponded to spring tide conditions. The measured parameters were water depth, spatially averaged velocity profile (based on measurements at four locations), water surface slope along a major flow direction, and bed shear stress. The flow at the site is considered fully

developed.

For the model run, we reduced the number of vertical layers to 5 compared to the previous section because of the shallow water depths. We imposed the measured water surface slope at the boundaries to drive the flow, where the water depths at the boundaries were adjusted to realize the same water depth at the center of the model domain as the measurement.

We used a value 1.0 for the drag coefficient ($C_D$) and 0.5 mm for the bottom roughness ($z_0$) based on the results in Yoshikai

et al. (2022a) (Table 2). As in the previous section, we changed the value of $\gamma$ in a reported range (0.8–1.6) with an interval of 0.1 to seek a value that produced the best fit with the measured velocity profile. Note that the TKE profile has not been measured in the field, thus it could not be validated.

We tested five different model configurations: $Rh$ model using the measured values for $a_{root}$ (actual $a$; black markers in Fig. 3b), $Rh$ model using the modeled $a_{root}$ (blue markers in Fig. 3b), cylinder model using two different cylinder densities

(sparse and dense; dashed lines in Fig. 3b), and a case without imposing the vegetation drag (no vegetation). Among these, the proposed framework (Fig. 1) was used for the case $Rh$ model using the modeled $a_{root}$ (the *Rhizophora* root module provided the $a_{root}$ in the simulation). We inputted the measured mean stem diameter ($D_{stem,ave}$) and tree density ($n_{tree}$) for the configuration using the $Rh$ model with modeled $a_{root}$. We set the sparse cylinder case based on Horstman et al. (2013) that suggested the use of vegetation geometry measured at a height of around 0.25 m for cylinder array approximation. We set the dense cylinder

array to produce an equivalent resistance to Manning's coefficient of 0.14 at a water depth of 0.5 m, a value often used to

represent the drag by mangroves (e.g., Zhang et al., 2012; Menéndez et al., 2020). For the case without vegetation, the bed shear stress is the main force to equate with the imposed pressure gradient.

**2.3 Sediment transport simulation**

We conducted the sediment transport simulation using the ROMS–CSTM coupled model implemented in COAWST adopting the same forcing conditions for flows in Bakhawan Ecopark during the ebb tide on September 11, 2018 as described in the previous section. We gave a vertically and temporally constant suspended sediment concentration (SSC) of 50 mg L$^{-1}$ to the upstream boundary condition, based on some modeling studies for mangrove forests and salt marshes (e.g., Mariotti and Fagerazzi., 2010; Horstman et al., 2015). Although flows simulated were of ebb tide, we assumed the SSC flux from the upstream boundary as sediment supply from water bodies such as creeks. The key parameters that characterize sediment properties are settling velocity ($w_s$, mm s$^{-1}$) and critical shear stress for erosion ($\tau_{cr}$, N m$^{-2}$), where $w_s$ controls the sediment deposition flux and $\tau_{cr}$ determines the threshold of bed shear stress ($\tau$) for sediment erosion initiation, above which the erosion flux linearly increases as $\tau$ increases (see Warner et al., 2008 for the details). We gave 0.1 mm s$^{-1}$ to $w_s$ and 0.1 N m$^{-2}$ to $\tau_{cr}$ based on the model configuration for sediment transport in a mangrove forest by Horstman et al. (2015). Table S2 summarizes the other sediment parameter values. Similar to many other numerical studies that examined sediment dynamics in coastal vegetated areas, we represented the impact of vegetation on sedimentation through the reduced bed shear stress that regulates sediment erosion (e.g., Zhang et al., 2019; Zhu et al., 2020; Breda et al., 2021; Zhang et al., 2022). We did not activate processes of elevation change and bed load in the simulation.

Similar to the previous section, we run the model with four different model configurations, which are *Rh* model using the modeled $a_{root}$, sparse cylinder model, dense cylinder model, and no vegetation to explore the impacts of different representations of *Rhizophora* mangroves in the model. We set a 200-m long transect located between the middle points of the south and north boundaries in the model domain. Then, we calculated three variables from each model output for a steady state flow and sediment condition–transect-mean bed shear stress, deposition fraction across the transect relative to the supplied sediments, and transect-mean sedimentation rate.

**3 Results**

**3.1 Comparison with a laboratory-based study**

Figure 4 shows a comparison of the modeled and measured vertical profiles of velocity (*u*) and TKE (*k*) normalized by cross-sectional mean velocity (*U*) for Exp 1 and 2, the conditions examined by Maza et al. (2017). The profile of normalized velocity was reasonably predicted by the *Rh* model (Fig. 4a, c), especially at the lower part of the root system (i.e., $z/HR_{max} <$ 0.6) in Exp 1 where the velocity was greatly attenuated compared to the upper part or above the root system (Fig. 4a). The higher values of the *γ* lead to more homogeneous velocity profiles because of the enhanced vertical momentum exchange by the elevated TKE, while the sensitivity to the varying *γ* was not significant. The *Rh* model also predicted well the overall trend





of the normalized TKE profile measured by Maza et al. (2017) for both Exp 1 and 2 by adjusting the value of $\gamma$ (Fig. 4b, d). Notably, the *Rh* model captured well the distinct vertical variations in TKE observed in Exp 1 when $\gamma = 1.5$ (Fig. 4b), while for Exp 2, the best fit was obtained when $\gamma = 0.9$ (Fig. 4d). Overall, $\gamma = 1.2$ produced the smallest total error of Exp 1 and 2

between the model and measured values. It under- and overestimated the TKE averaged over the measurement section by about 20 and 40 % for Exp 1 and 2, respectively (Fig. 4b and d), which is generally a fairly good agreement for predicting TKE.

In contrast to the *Rh* model, the cylinder model predicted the nearly uniform vertical profile of velocity except the region close to the bed both for Exp 1 and 2, and largely deviated from the measurements (Fig. 4a and c). The TKE predicted by the

cylinder model also showed the nearly uniform vertical profile (Fig. 4b and d). While the cylinder model showed comparable TKE with the *Rh* model at the lower part of the root system (i.e., $z/HR_{max} < 0.4$) for both cases, it showed the significantly smaller TKE at the upper region from the *Rh* model and the measurement.

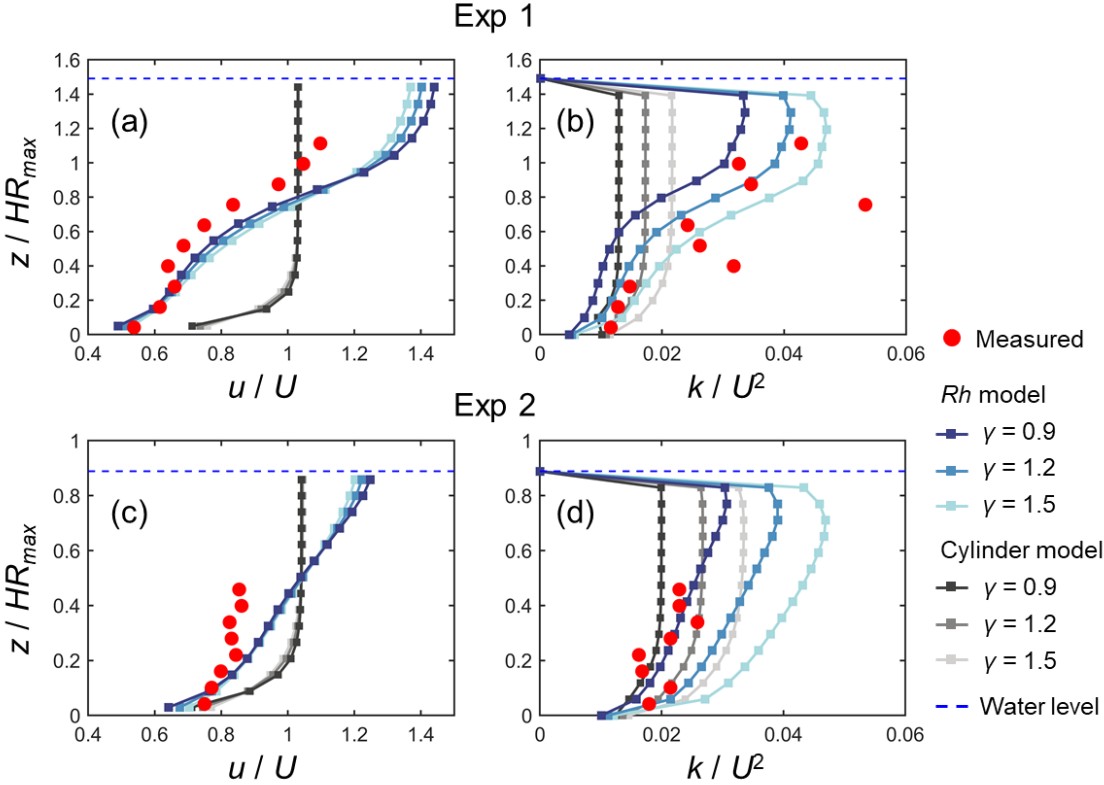

Figure 4. Comparison of the vertical profiles of (temporally and spatially averaged) velocity ($u$) and turbulent kinetic energy

($k$) normalized by the cross-sectional mean velocity ($U$) predicted by the COAWST with different model configurations (*Rh*





model and cylinder model) and with different $\gamma$ values, and measurement by Maza et al. (2017) for (a, b) Exp 1 and (c, d) Exp 2.

## 3.2 Comparison with a field-based study

Figure 5 shows the comparison of modeled velocity profiles with measurements by Yoshikai et al. (2022a) for some

selected tidal phases in a *Rhizophora* mangrove forest (Bakhawan Ecopark). The *Rh* model using the measured profile of root projected area density (actual $a_{root}$) predicted well the overall trend of measured velocity profiles in various tidal phases (Fig. 5a). However, the model seemed to have underestimated the velocity attenuation from the surface to the bottom, which resulted in slightly higher near-bottom velocity and/or lower near-surface velocity compared to the measurement. Here, the value of $\gamma$ was chosen as 0.8 from the range 0.8–1.6 (Table 2), which produced the best fit with the measured velocity profile. The *Rh*

model using the modeled $a_{root}$ provided by the *Rhizophora* root module showed a comparable performance with the use of actual $a_{root}$ in predicting the velocity profile (Fig. 5a). However, although not significant, the use of modeled $a_{root}$ tended to further underestimate the velocity attenuation from the surface to the bottom due to the underestimation of $a_{root}$ near the bed by the *Rh*-root model (Fig. 3b).

The cylinder model with sparse arrays showed comparable velocities with measurements near the water surface, but

significantly overestimated the velocities near the bed (Fig. 5b). Alternatively, the dense arrays showed comparable velocities near the bed, but significantly underestimated the velocities near the water surface (Fig. 5c). The condition without vegetation drag lead to a large overestimation of the velocities, approximately 3–4 times larger than the measurements (Fig. 5d).



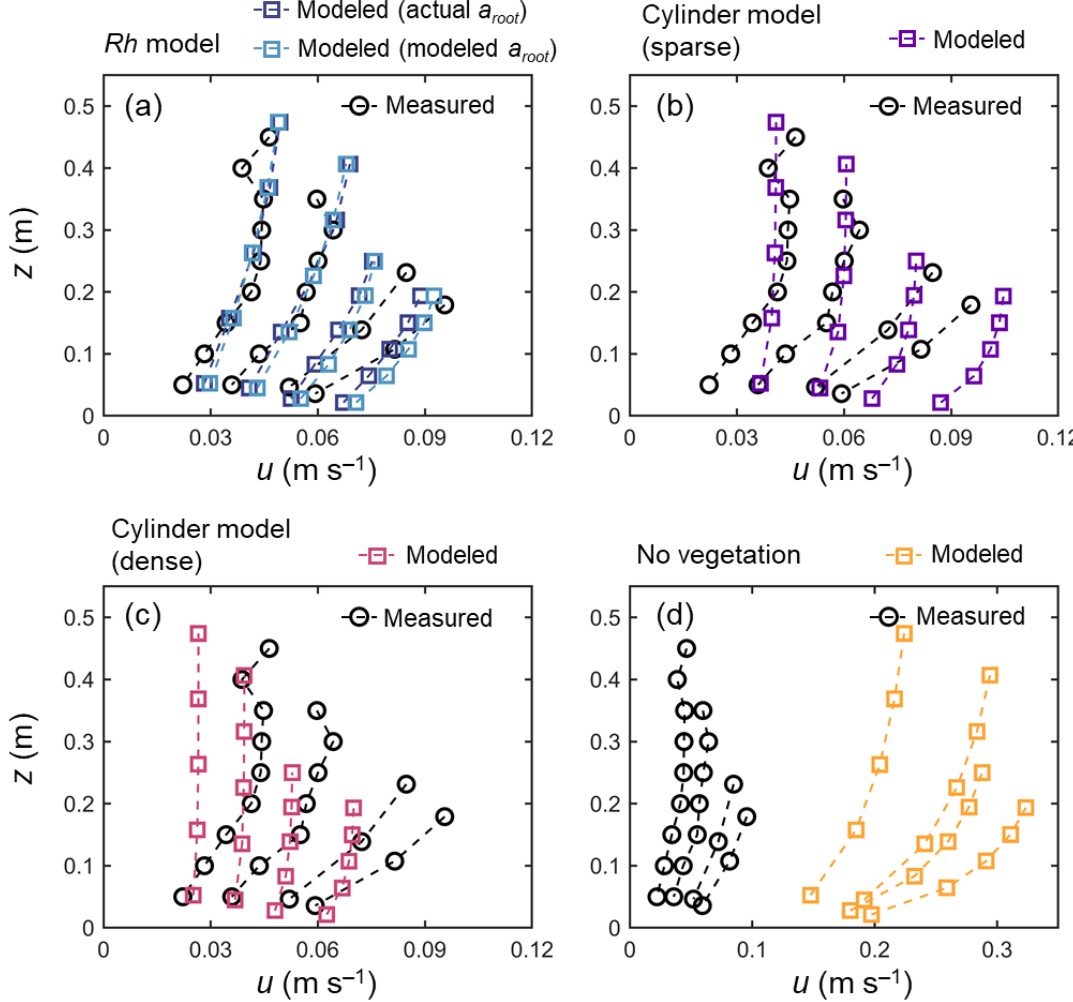

Figure 5. Comparison of the vertical profiles of velocity ($u$) predicted by the COAWST employing (a) *Rh* model using actual
and modeled root projected area density profile ($a_{root}$), (b) cylinder model with sparse and (c) dense array, and (d) without
imposing vegetation drag (no vegetation), and measurement by Yoshikai et al. (2022a) for some selected tidal phases during
the measurement period.

A fairly good reproduction of tidal flows by the *Rh* model can also be seen in the agreement with the measurement for
the time-series of channel-mean velocity ($U$), (spatially averaged) velocity at $z = 0.05$ m ($u_{bottom}$), and bed shear stress ($\tau_{bed}$)
during the 2-days measurement period (Fig. 6). Note that we estimated the model prediction of velocity at $z = 0.05$ m from
linear interpolation of velocities computed at adjacent vertical layers. The $u_{bottom}$ was generally overestimated by about 15 %
(Fig. 6c, g), as also seen in Fig. 5a. As a result, the $\tau_{bed}$ was overestimated by about 30 % by the model, which is still a





reasonable agreement (Fig. 6d, h). As demonstrated in Fig. 5a, the *Rh* model employing the modeled $a_{root}$ also showed a comparable performance for the time-series data (Fig. 6).

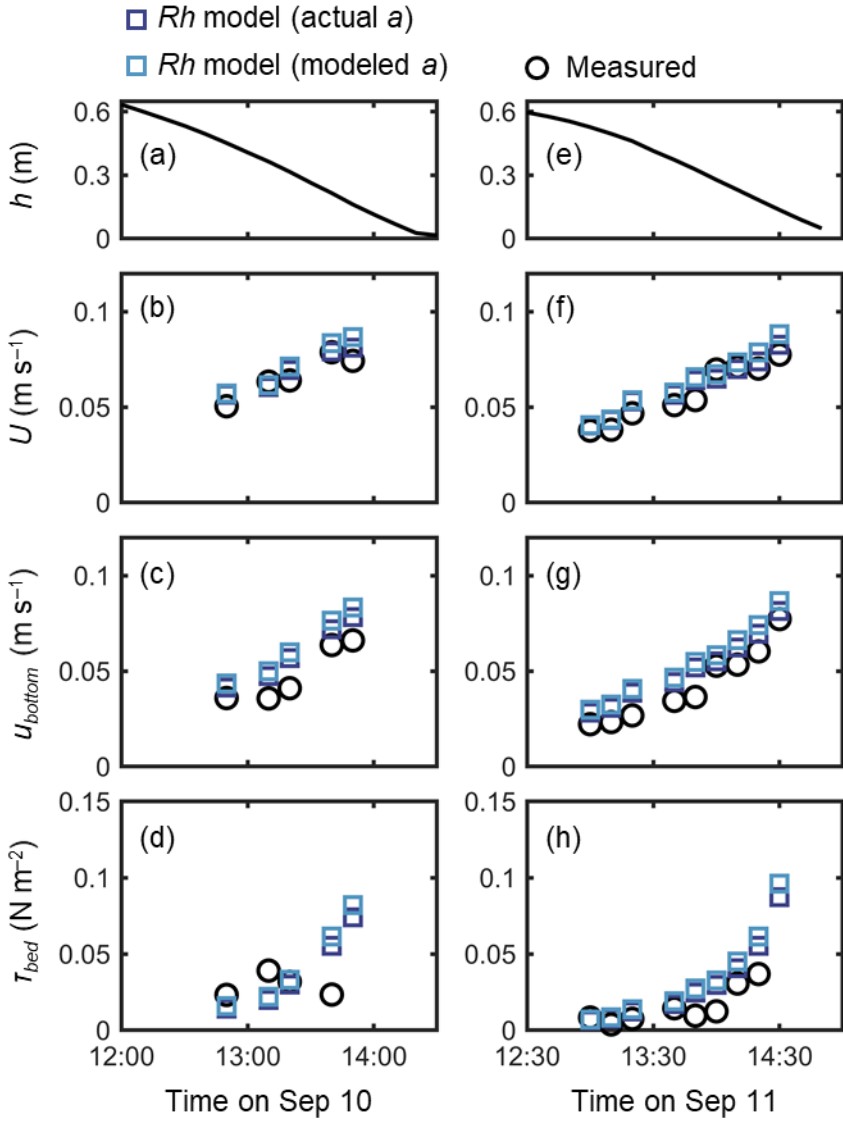

Figure 6. Time-series of (a, e) measured water depth (*h*), measured and predicted (b, f) cross-sectional mean velocity (*U*), (c, g) (spatially averaged) velocity at $z = 0.05$ m, and (d, h) bed shear stress ($\tau_{bed}$) during the two-days measurement in Bakhawan Ecopark. The measured values are from Yoshikai et al. (2022a) and the predicted values are obtained through the COAWST employing the *Rh* model using actual and modeled root projected area density profile ($a_{root}$).




The cylinder model with sparse array led to a significant overestimation trend of the $U$, $u_{bottom}$, and $\tau_{bed}$ over the tidal phases especially when the water depth decreased (Fig. 7). The cylinder model with dense array led to the underestimation of $U$ most of the tidal phases but showed an agreement with the measurement for $u_{bottom}$ and $\tau_{bed}$ (Fig. 7). The model without imposing vegetation drag led to large overestimation of these parameters over the tidal phases (Fig. S1), similar to the result shown in Fig. 5d.

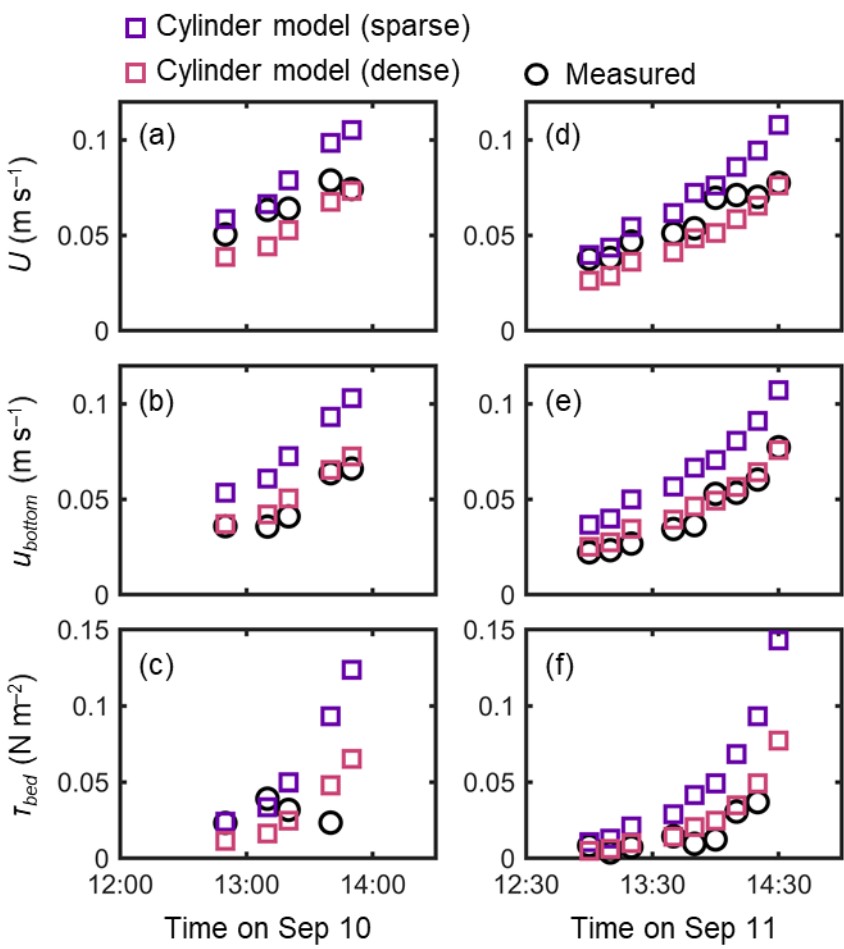

Figure 7. Time-series of measured and predicted (a, d) cross-sectional mean velocity ($U$), (b, e) (spatially averaged) velocity at $z = 0.05$ m, and (c, f) bed shear stress ($\tau_{bed}$) during the two-days measurement in Bakhawan Ecopark. The measured values are from Yoshikai et al. (2022a) and the predicted values are obtained through the COAWST employing the cylinder model with sparse and dense arrays.



## 3.3 Sediment transport simulation

Figure 8 shows the results of the sediment transport simulation conducted for Bakhawan Ecopark on September 11$^{th}$ using the different model configurations. The bed shear stress, $\tau_{bed}$, showed some variations across the transect (as shown in the error bars in Fig. 8a) due to the velocity variations attributed to the water depth variations. The no vegetation case, wherein $\tau_{bed}$ largely exceeded the critical shear stress ($\tau_{cr}$), resulted in the loss of sediments (negative sedimentation rate) over the tidal phase. In contrast, the cases $Rh$ model and cylinder model showed $\tau_{bed}$ lower than the critical shear stress ($\tau_{cr}$) in most of the tidal phases. The cylinder model with a sparse array showed the highest $\tau_{bed}$ among the vegetated cases, wherein the last two $\tau_{bed}$ (i.e., at 14:20 and 14:30) exceeded the $\tau_{cr}$. This resulted in the lowered sediment deposition fraction and sedimentation rate (Fig. 8b and c). Specifically, the sedimentation rate became negative at 14:30 due to the excess flux of erosion than deposition at this condition. The case using the $Rh$ model showed the next highest $\tau_{bed}$, wherein some portion of the transect exceeded the $\tau_{cr}$ at 14:30 (Fig. 8a). Although this reduced the sediment deposition fraction and sedimentation rate, they remained at positive values (Fig. 8b and c). Overall, the $Rh$ model showed similar deposition fractions and sedimentation rates with the cylinder model with a sparse array except for the last two predictions. The cylinder model with a dense array showed the lowest $\tau_{cr}$ (Fig. 8a). The significantly reduced flow velocity (as shown in Fig. 7a, d) induced the high deposition fraction (about 80 %) over the tidal phase while the waters were passing the 200-m transect, which was evidently higher than the other cases (Fig. 8b). However, the reduced flow velocity, which reduced the sediment supply to the model domain, led to the lower sedimentation rates compared to the $Rh$ model and cylinder model with a sparse array in most of the tidal phases (Fig. 8c). As a result, the $Rh$ model showed the highest tidally averaged sedimentation rate (note that flood tide is not included), 18 % and 11 % higher than the cylinder model with sparse and dense arrays, respectively. Here we estimated the sedimentation rate at 13:30, the time when the data to run the model is not available, by taking the average at 13:20 and 13:40 for calculating the tidally-averaged values.





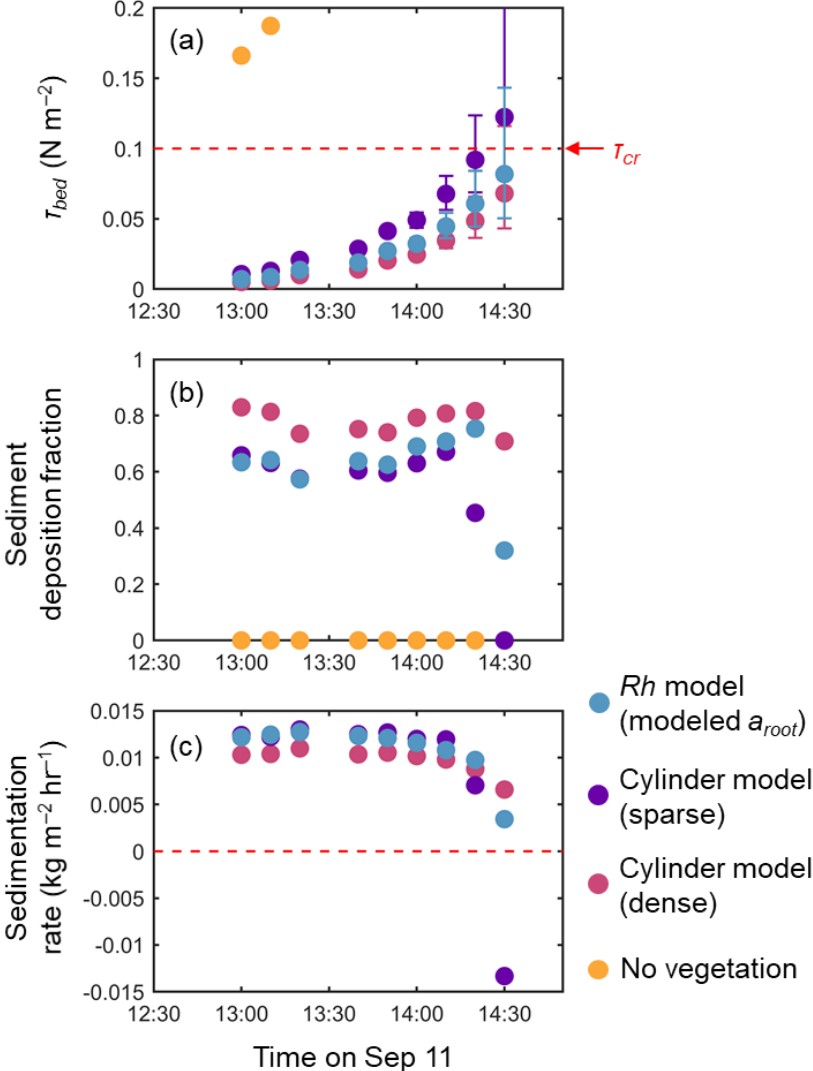

Figure 8. (a) Transect-mean bed shear stress ($\tau_{bed}$), (b) sediment deposition fraction across the transect relative to the supplied sediments, and (c) transect-mean sedimentation rate simulated using hydrodynamic conditions in Bakhawan Ecopark on Sep 11 by the COAWST using the *Rh* model using the modeled root projected area density profile ($a_{root}$), cylinder model with 405 sparse and dense arrays, and without imposing the vegetation drag (no vegetation). Error bars in panel (a) show the highest and lowest bed shear stress across the transect. For the no vegetation case, the mean bed shear stress exceeded the axis limit after the first two predictions (see Fig. S1c, f). Also, the sediment deposition fraction in the no vegetation case was always zero where the marker of the last prediction was hidden by the cylinder model with sparse array. The sedimentation rate in the no vegetation case was from –0.05 to –0.24 kg m$^{-2}$ s$^{-1}$, which likewise exceeded the lower axis limit. The red dashed lines in 410 panels (a) and (c) refer to the critical shear stress ($\tau_{cr}$) and zero value, respectively.




## 4 Discussion

### 4.1 Model performance

Due to the general lack of information on the vertically varying projected area of the complicated root systems, the drag by *Rhizophora* mangroves has been represented by the increased Manning's roughness coefficient values (e.g., Zhang et al.,

2012) or an array of cylinders with arbitrary cylinder density (Horstman et al., 2013; Xie et al., 2020) in hydrodynamic models with 2D configuration. We evaluated these drag parameterizations using the cylinder array approximation in 3D configuration (dashed lines in Fig. 3). Consistent with previous studies (Liu et al., 2008; King et al., 2012), the cylinder array approximations showed the vertically uniform velocity and TKE profile except near the bed, which largely deviated from the measurements (Figs. 4 and 5). Moreover, for the tidal flows with changing water depth, the two different cylinder array configurations (sparse

and dense) failed to capture the velocity changes over the tidal phases (Fig. 7) due to the inability to capture the changes in submerged vegetation projected area of *Rhizophora* mangroves.

We proposed a new drag and turbulence model for flows in *Rhizophora* mangrove forests that works on the 3D hydrodynamic model, ROMS, and implemented in the COAWST. The model explicitly accounts for the vertically varying projected area of the root systems for drag force and TKE production in 3D configuration. In addition, the model accounts for

the two different length-scales of wakes (roots- and stem-generated wakes) in the turbulence closure model ($k$–$\varepsilon$ model in this study), an aspect none of the modeling studies has examined yet (e.g., López and García, 2001; King et al., 2012). With the relatively simple modifications made to the equations introduced by Beudin et al. (2017) (Eqs. 1, 4–6), our results showed significantly improved reproducibility of ROMS for the vertical profiles of velocity and TKE, and velocity changes over the tidal phases in *Rhizophora* mangrove forests (Figs. 4, 5, and 6). The new model also reasonably predicted the bed shear stress

together with these parameters (Fig. 6d and h). Although some studies have accounted for the vertically varying vegetation projected area in hydrodynamic models for salt marshes (Temmerman et al., 2005) or mangrove forests with *Rhizophora* stands (Horstman et al., 2013, 2015), the efficacy of accounting for the vegetation 3D structures in the model has not been demonstrated. Overall, this is the first modeling study to introduce the realistic representation of the influences of *Rhizophora* mangrove morphological structures on the flow that was validated with existing data. The good performance of the model in

both the model- and real-*Rhizophora* mangrove forests having a range of vegetation complexity (Fig. 3) suggests the model's general applicability to *Rhizophora* mangrove forests worldwide.

The laboratory-based study of Maza et al. (2017) provided valuable data for evaluating the new model for TKE in a *Rhizophora* mangrove forest, which is currently unavailable from field-based studies. They observed the elevated TKE at the upper root zone and above the root zone ($z/HR_{max} > 0.5$; Fig. 4b). Maza et al. (2017) discussed the TKE production by shear

($P_s$ in Eq. 2) as one of the main reasons of the elevated TKE. However, we found that the different dominance of the root- and stem-generated wakes over the depth can explain it, such that the lower root zone dominated by root-generated wakes with length-scale set as root diameter (0.038 m; Table 2) resulted in a higher dissipation rate (Eq. 6a), thereby lower TKE; and the



higher root zone dominated by stem-generated wakes with length-scale set as stem diameter (0.2 m; Table 2) resulted in a lower dissipation rate (Eq. 6b), thereby higher TKE; and the nonlinear transition between them (Fig. 4b). This result is similar

to the observation by Xu and Nepf (2020) that found vertically varying turbulence integral length-scale in a canopy of a salt marsh plant *Typha*. Without accounting for the two different length-scales, the model failed to reproduce the TKE profile while the velocity profile remained similar, suggesting the minor contribution of shear production to TKE (results not shown). The model also predicted the gradually increasing TKE upwards in the lower root zone ($z/HR_{max} < 0.5$), which is consistent with the measurement (Fig. 4b, d). While the model showed good reproducibility of the TKE profile, it should be noted that different

$\gamma$ values produced the best fit with the measurement for Exp1 and Exp2 (Fig. 4b, d). At this moment, the exact explanation for this observation is yet to be determined, whether it can be attributed to measurement uncertainty or processes that were not represented in the model. Further research on the turbulence structures in *Rhizophora* mangrove forests is needed. Unlike the TKE profiles predicted for the model mangrove forest, the TKE predicted for the field mangrove forest (Bakhawan Ecopark), which has much higher vegetation complexity (higher $a$; Fig. 3), showed nearly uniform vertical profiles (Fig. S2), results we

cannot validate at this moment. Field studies on turbulence structures are likewise needed in this sense.

From the results shown, we highlighted the importance of accounting for the vertically varying projected area of the root systems with 3D configuration for capturing the flow structures (Figs. 4–7). The model predictability is therefore dependent on the root projected area, which is typically unknown and labor-intensive to measure (Yoshikai et al., 2021). For the practical use of the model, we proposed a model framework (Fig. 1) leveraging an empirical model for the *Rhizophora* root system (*Rh-*

root model) with parameterization of subgrid-scale tree variations (Fig. 2), which we implemented in COAWST. The simulation for flows in Bakhawan Ecopark using the modeled $a_{root}$ provided by the *Rhizophora* root module showed almost identical results with the one using the measured $a_{root}$ in the field (Figs. 5 and 6). This indicates the worth of the model framework for practically and accurately predicting the flows in real *Rhizophora* mangrove forests. The grid-scale parameters required are mean stem diameter ($D_{stem,ave}$) and tree density ($n_{tree}$), which are basic information collected during tree census

surveys (Simard et al., 2019; Suwa et al., 2021). Although the process of collecting these spatial data is out of the scope of the study, we expect that even remotely sensed data such as airborne LiDAR (Jucker et al., 2017; Dai et al., 2018) or UAV optical imagery (Otero et al., 2018), which can detect basic tree features (e.g., tree height and crown width) that have strong relationship with stem diameter (Jucker et al., 2017; Azman et al., 2021), can provide such information effectively. Obtaining the root scaling parameters requires field surveys (Fig. 1); however, these parameters can be relatively easily obtained by

sampling 10–20 trees at the site (see Yoshikai et al., 2021, 2022a for the procedure). The collection of these data is far less exhaustive than extensively measuring the vertical profile of $a_{root}$ in the area of interest as done in Horstman et al. (2015). This study thus offers the first framework of numerical modeling which can be readily applied to *Rhizophora* mangrove forests in the field.





**4.2 Implication for sediment transport in *Rhizophora* mangrove forest**

Using the improved representation of hydrodynamics in *Rhizophora* mangrove forests, we explored how *Rhizophora* mangroves impact sedimentation during normal conditions (from high to low tide in this study; Fig. 8). The water surface slope that drives the water flow was identical among the four simulation cases (*Rh* model, sparse and dense cylinder arrays, no vegetation). Although the hydrodynamic conditions were of ebb tide, the different sedimentation rates among the cases thus signify how the different vegetation configuration could impact the water flow, hence sediment supply, from the channel to

the forest, and the deposition and retention of sediments (Xu et al., 2022).

The consistent negative sedimentation rates in the no vegetation case in the simulation period due to excess erosion over deposition suggests that the fine sediments typically found in mangrove forests are not likely deposited in non-vegetated areas, i.e., creeks (Fig. 8c; Willemsen et al., 2016). In contrast, the vegetated cases showed the positive sedimentation rates with high deposition fractions (more than 60 %) in the 200-m long transect during most of the tidal phases. This suggests the significance

of vegetation in reducing bed shear stress and promoting the deposition and retention of fine sediments.

Vegetation has the competing effects of supply and retention on sedimentation, primarily controlled by vegetation density (Nardin and Edmonds, 2014; Zhang et al., 2019; Olliver et al., 2020; Xu et al., 2022). The effects can be seen in the results of sparse and dense cylinder arrays, that supply higher/lower amount of water and sediments to the forest, but less/more effectively regulate sediment erosion (Fig. 8). As a result, the tidally averaged sedimentation rates were comparable between

the sparse and dense cylinder cases in this study ($9.3 \times 10^{-3}$ and $9.9 \times 10^{-3}$ kg m$^{-2}$ hr$^{-1}$, respectively). Interestingly, the *Rh* model showed a tidally averaged sedimentation rate significantly higher than the both cylinder cases ($11.0 \times 10^{-3}$ kg m$^{-2}$ hr$^{-1}$). This higher efficiency in sedimentation shown by the *Rh* model is related to the vertical variations in the projected area of *Rhizophora* mangrove root systems (Fig. 3b). The relatively low projected area at the higher portion allowed the higher water and sediment influxes to the forest, which are comparable to the sparse cylinder case when $h > 0.3$ m (Fig. 8c; see Fig. 6e for

$h$). The high projected area at the lower portion, on the other hand, effectively reduced the bed shear stress that is comparable to the dense cylinder case (Fig. 8a) and prevented erosion. Such features of *Rhizophora* mangroves that take advantage of sparse and dense cylinder arrays in terms of sediment supply and retention, respectively, contributed to the highest sedimentation rate. This may explain overall observed trends of high sediment accretion rate and trapping of large amount of carbon originated from outer systems in the soil of mangrove forests with *Rhizophora* stands (Krauss et al., 2003; Suello et al.,

2022). This study thus offered a new insight that in addition to the vegetation density, the vertical variations in vegetation projected area that modulate the velocity profile could be a major factor that control the sedimentation rate in vegetated areas such as in *Rhizophora* mangrove forests.

It should be noted, however, that the prediction of sedimentation rate largely depends on the boundary conditions of the SSC, which is influenced by e.g., geophysical settings, tides, and season. Hence, the tidally averaged sedimentation rates

demonstrated in this study using the constant SSC may not always reflect the actual conditions. Future studies on field



observations and model applications are thus needed to evaluate the actual sedimentary processes in *Rhizophora* mangrove forests.

We also note that the sediment erosion rate is determined solely by bed shear stress in the present hydrodynamic-sediment transport model, similar to most modeling studies (e.g., Zhang et al., 2022). Several recent experimental studies have shown
that the turbulence generated by vegetation could contribute to sediment erosion, and that turbulence represented by TKE may be a better predictor of erosion rate than bed shear stress (Tinoco and Coco, 2016; Yang and Nepf, 2018; Liu et al., 2021), a process not considered in this study. In this regard, the sediment retention function by vegetation may have been somewhat overestimated in the simulation, which needs further investigation. Yet, accounting for the impact of vegetation-generated turbulence on erosion in hydrodynamic models is not simple. In hydrodynamic models, TKE is usually computed at the vertical
layer interfaces, where the near-bed TKE (at the sediment-water interface) is solely given from bed shear stress, and the contribution of vegetation to TKE is accounted for only at the upper interfaces (Warner et al., 2005; King et al., 2012). Further modeling works are needed to resolve this aspect. At present, the insights into the effects of turbulence on sediment erosion in *Rhizophora* mangrove forests are very limited, necessitating further laboratory- and field-based studies.

### 4.3 Further model improvement

This study has not examined processes such as vertical mixing and longitudinal dispersal in *Rhizophora* mangrove forests due to the lack of data, which are also relevant to sediment and other substance transport (Lightbody and Nepf, 2006; Xu and Nepf, 2021). Nevertheless, given the improved ability to predict the vertical profile of velocity and TKE in addition to the depth-averaged velocity, the model may better represent these processes than other models. Future experimental and field studies are needed to support and improve the model ability.

In order to predict the long-term geomorphic evolution of mangrove forests, the interactive feedback of vegetation-flow-sediment needs to be precisely simulated (van Maanen et al., 2015; Rodríguez et al., 2017; Xie et al., 2020). This process involves long-term changes in root structure complexity in accordance with forest growth/development–a process poorly represented in previous studies in the case of *Rhizophora* mangrove forests. An advantage of the proposed model is that the root structures of *Rhizophora* mangroves are allometrically predicted in the hydrodynamic model from the basic forest
structural variables–mean stem diameter and tree density, of which long-term dynamics can now be predicted using dynamic vegetation models for mangroves (e.g., Yoshikai et al., 2022b). The coupling of the hydrodynamic-sediment transport model and the dynamic vegetation model is one of the next challenges that will advance our understanding of the long-term sedimentary processes for the evaluation of the physical aspect of mangroves' ecosystem services and vulnerability.

### 5 Concluding remarks

This manuscript presented a new model to represent the impacts of *Rhizophora* mangroves on flow and sediment transport implemented in the COAWST. We showed that compared to the conventional cylinder array approximation, the new model





that explicitly accounts for the three-dimensional root structures, and the two potential length-scales of vegetation-generated turbulence significantly improves the prediction of velocity and TKE in *Rhizophora* mangrove forests. Through the numerical experiment on sediment transport, the model suggested the high efficiency of *Rhizophora* mangroves on sedimentation which
allows relatively high sediment supply, but effectively regulate sediment erosion by reduced bed shear stress, processes which cannot be captured by the cylinder array approximation. Thus, accounting for the realistic morphological structures of *Rhizophora* mangroves in the hydrodynamic model with three-dimensional configuration is important for simulating the flow and sediment transport in *Rhizophora* mangrove forests. While obtaining the information on the root structures in the field of model application could be challenging, the new model is now feasible in its application to the field due to the incorporation
of the empirical model for *Rhizophora* root structures to the COAWST. While there are some important processes to be addressed to further improve the model (e.g., the effect of vegetation-generated turbulence on sediment erosion), the model developed here may serve as a fundamental tool to advance our understanding of the sedimentary processes in *Rhizophora* mangrove forests, and thus the mangroves' ecosystem services and vulnerability.

**Code and data availability**

The model codes, input data, and run scripts are available at https://zenodo.org/record/7353835#.Y4BWYJrP1D8 (Yoshikai, 2022).

**Author contributions**

MY, TN, and KN designed the study and developed the model proposed in the manuscript. MY made the necessary modifications to the model code of the COAWST and performed the analyses. TN and KN contributed to result interpretation.
MY wrote the manuscript, and all the authors contributed reviewing and editing.

**Competing interests**

The authors declare that they have no conflict of interests.

**Acknowledgements**

The authors gratefully acknowledge the Japan International Cooperation Agency (JICA) and Japan Science and Technology
Agency (JST) through the Science and Technology Research Partnership for Sustainable Development Program (SATREPS) for financially supporting the project Comprehensive Assessment and Conservation of Blue Carbon Ecosystems ad their Services in the Coral Triangle (BlueCARES). We thank Dr. Charissa Ferrera for providing language help.



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
