# Peer review of "Representing the impact of *Rhizophora* mangroves on flow in a hydrodynamic model (COAWST\_rh v1.0): the importance of threedimensional root system structures"

_EGUsphere, 2022_

## Author Comment (AC1)

Dear editor and reviewers,

We thank the reviewers for taking the time to review our manuscript and providing constructive comments. We apologize for the delay in submitting our responses to the comments.

Below, we address the reviewers' comments point by point. In the response, black- and blue-colored characters denote reviewers' comments and our responses, respectively.

**Response to Reviewer #1**

**Reviewer [1.1]:**
Yoshikai et al present a novel implementation of a mangrove root model in the ROMS data package that has the potential to be very useful for the wider scientific community. They show that the model predicts flow velocities and turbulent kinetic energy in a more refined way in the vertical and that this matches qualitatively better with the presented data from a flume experiment and the field. Although I think that the model is very useful and can advance the scientific field, I have major concerns with the application of the model to the sediment transport predictions that links directly to large parts of the discussion. In addition, I missed more detail on the data that has been used. Finally, to help lift this contribution from a presentation of the model, I suggest to apply the model to a case study with available sediment transport rates and potentially morphodynamic change to show that those predictions are improved using the presented model. I outline my concerns below:

**Response [1.1]:**
We thank the reviewer for the thorough assessment and constructive comments on our manuscript.

Regarding the reviewer's concern about the model application to sediment transport, we admit that the model efficacy on sediment transport is not well demonstrated in this study compared to the flow structures in the mangrove forests. While the reviewer's suggestion is interesting and an important topic to address, we would like to note that model application to sediment transport in mangrove forests and its evaluation is currently constrained by data availability. Even for the prediction of flows, comprehensive data sets such as hydrodynamics (e.g., water depth and flow velocity), vegetation parameters (both stems and roots), as well as boundary conditions of the flow (e.g., water level gradient) are required to drive and evaluate the developed model but are rarely available especially in the field. The application and evaluation for the sediment transport require additional data such as sediment properties (e.g., grain size), suspended sediment concentration, sedimentation rate, and morphodynamics. To our knowledge, the data set that satisfies such requirement is currently unavailable from both flume- and field-based studies. Hence, at this moment, we are not able to address the reviewer's suggestion regarding the application of the model to a case study

with available sediment transport rates and morphodynamic change to demonstrate the efficacy of the presented model.

Our intention of the sediment transport simulation performed in this study was to demonstrate how the flow field created by the *Rhizophora* mangroves may change the sedimentation rates using the new model. However, given the current limitation to show the model's efficacy on sediment transport and that the main contribution of the study is the realization of the realistic flow simulation in *Rhizophora* mangrove forests, we would like to remove the results and discussion on the sediment transport simulation in the revised manuscript, as suggested in the comment [1.5]. This is also suggested by Reviewer #2 (please see the comment [2.7]).

We have provided our responses to the individual comments below.

**Reviewer [1.2]:**

1) To understand where the data comes from I suggest to add maps of both the study area and the model grid with flow velocities as well as the set-up of the flume experiments/model iof the flume experiments. It is unclear where the measurements have been taken (unvegetated vs. within the root system, close to tidal channels etc) and how exactly the model looks to allow to understand the results and to reproduce the study. In addition the data used for validation should be presented in the supplementary.

**Response [1.2]:**

We have provided in Fig. R1 of this document the maps of the field measurement of Yoshikai et al. (2022) used for the model application in this study, and in Fig. R2 the schematic of the model grid used for model testing against the flume experiment and field measurement. The set-up of the flume experiment has been described in detail in Maza et al. (2017). Because the reuse of the figures in Maza et al. (2017) in this manuscript would cost us a substantial amount of payment to the publisher of the original article, we would like to refer the reviewers and readers to their original article. We have also provided in Tables R1–3 below the data used for model validation. Furthermore, for a better grasp of the measurements of Maza et al. (2017) and Yoshikai et al. (2022) and the model setting, we have provided Table R4 summarizing the measured flow variables, the setting of model forcing, and the target variables to reproduce for each application.

To avoid any confusion regarding the model setting, we would like to note that the model was tested with the model grid assuming a schematized mangrove forest (Fig. R2) as described in L216–225, not with a grid representing the actual geometric/topographic conditions of the flume/field. Also, bed elevation and vegetation parameters in the grid were set uniformly over the model domain as described in L217–218. This simplification of the model setting is deemed reasonable given the (approximately, in the case of the field mangrove forest) spatially uniform vegetation distribution and the well-developed flow conditions at the flow measurement location in both Maza et al. (2017) and Yoshikai et al.

(2022) where the dependence of flow structures on the proximity to the forest leading edge is diminished. The flow in the model was driven by a water level gradient imposed between the open boundaries (Fig. R2; L218–220). We then created a steady state of flow in the model and compared the simulated flow at the monitoring point in the model domain (Fig. R2) with the data (L222–223). This means that for the model application to the field mangrove forest, the actual time-series of the flow has not been reproduced; rather, steady states of flow were created for each flow measurement.

As an action for manuscript revision, we will include Figs. R1–2 and Tables R1–4 in the Supporting Information and include any missing descriptions on the model settings in the main text.

[Figure]

Figure R1. (a) Satellite image (Google Earth) of the study site of Yoshikai et al. (2022) – Bakhawan Ecopark (red box indicates the area of panel "b"), (b) locations of transect A–B across which the water level gradient was measured together with the hydrodynamic parameters around the reference tree, (c) top view of LiDAR point clouds around the reference tree with information on the locations of trees whose morphological structures were measured, velocity profiling, and deployed sensors (velocity sensor: electromagnetic velocity meter deployed near the bottom; ADV: Acoustic Doppler Velocimeter deployed to estimate the bed shear stress). It has been shown in Yoshikai et al. (2022) that the average of the velocity measured at the four locations well represents the spatially-averaged values. The point clouds shown were cropped at heights between 0.1–1.7 m for better visualization of the root systems. Figures are modified from Yoshikai et al. (2022).

[Figure]

Figure R2. Model grid (40 × 40 with 5 m horizontal resolution) used for testing the model against laboratory-based and field-based studies. The red circle indicates the location of the monitoring point at which the simulated flow variables were compared with the measured data. Bed elevation and vegetation parameters were set uniformly over the model domain.

Table R1. Data from the flume experiments of Maza et al. (2017) that were used for the model validation in Figure 4. The values of geometric and flow parameters were converted from the scale in the flume to the real scale. The velocity ($u$) and turbulent kinetic energy ($k$) were taken by averaging the measurements at five lateral positions (ADV3p1–p5; see Fig. 5 of Maza et al., 2017) in the model mangrove forest where the flows were fully developed, which were taken as spatially-averaged values in the mangrove forest. $HR_{max}$: maximum root height, $h$: water depth, $U$: cross-sectional mean velocity, $z$: height above the bed.

| Experiment # | $HR_{max}$ (m) | $h$ (m) | $U$ (m s$^{-1}$) | $z$ (m) | $u/U$ | $k/U^2$ |
|---|---|---|---|---|---|---|
| Exp 1 | 2.016 | 3.0 | 0.31 | 0.08 | 0.54 | 0.012 |
| | | | | 0.32 | 0.62 | 0.013 |
| | | | | 0.56 | 0.66 | 0.015 |
| | | | | 0.80 | 0.64 | 0.032 |
| | | | | 1.04 | 0.69 | 0.026 |
| | | | | 1.28 | 0.75 | 0.024 |
| | | | | 1.52 | 0.84 | 0.053 |
| | | | | 1.76 | 0.97 | 0.035 |
| | | | | 2.00 | 1.05 | 0.033 |
| | | | | 2.24 | 1.10 | 0.043 |
| Exp 2 | 2.016 | 1.79 | 0.58 | 0.08 | 0.75 | 0.018 |

| | | | | 0.20 | 0.77 | 0.021 |
|---|---|---|---|---|---|---|
| | | | | 0.32 | 0.80 | 0.017 |
| | | | | 0.44 | 0.84 | 0.016 |
| | | | | 0.56 | 0.83 | 0.021 |
| | | | | 0.68 | 0.83 | 0.026 |
| | | | | 0.80 | 0.86 | 0.023 |
| | | | | 0.92 | 0.85 | 0.023 |

Table S2. Data from field measurements of Yoshikai et al. (2022) that were used for the model validation in Figure 5. The velocity ($u$) was obtained by averaging the measurements at four locations around the reference tree shown in Fig. R1c which was taken as spatially-averaged values in the mangrove forest.

| Local time | $h$ (m) | $z$ (m) | $u$ (m s$^{-1}$) |
|---|---|---|---|
| 2018/9/10 12:50 | 0.45 | 0.35 | 0.060 |
| | | 0.30 | 0.064 |
| | | 0.25 | 0.060 |
| | | 0.20 | 0.057 |
| | | 0.15 | 0.055 |
| | | 0.10 | 0.044 |
| | | 0.05 | 0.036 |
| 2018/9/10 13:40 | 0.21 | 0.18 | 0.096 |
| | | 0.11 | 0.082 |
| | | 0.04 | 0.059 |
| 2018/9/11 13:00 | 0.53 | 0.45 | 0.046 |
| | | 0.40 | 0.039 |
| | | 0.35 | 0.045 |
| | | 0.30 | 0.044 |
| | | 0.25 | 0.044 |
| | | 0.20 | 0.041 |
| | | 0.15 | 0.034 |
| | | 0.10 | 0.028 |
| | | 0.05 | 0.022 |
| 2018/9/11 14:00 | 0.28 | 0.23 | 0.085 |
| | | 0.14 | 0.072 |
| | | 0.05 | 0.052 |

Table R3. Data from field measurements of Yoshikai et al. (2022) that were used for the model validation in Figures 6 and 7. The $\Delta\eta$ is the water level difference imposed across the open boundaries in the model (see Fig. R2), $h$ is the water depth, $U$ is the cross-sectional mean flow velocity, $u_{bottom}$ is the spatially-averaged velocity at $z = 0.05$ m, and $\tau_{bed}$ is the bed shear stress.

| Local time | $\Delta\eta$ (m) | $h$ (m) | $U$ (m s$^{-1}$) | $u_{bottom}$ (m s$^{-1}$) | $\tau_{bed}$ (N m$^{-2}$) |
|---|---|---|---|---|---|
| 2018/09/10 12:50 | 0.0143 | 0.45 | 0.050 | 0.036 | 0.023 |
| 2018/09/10 13:10 | 0.0189 | 0.36 | 0.063 | 0.036 | 0.039 |
| 2018/09/10 13:20 | 0.0273 | 0.32 | 0.064 | 0.041 | 0.032 |

| 2018/09/10 13:40 | 0.0462 | 0.21 | 0.079 | 0.064 | 0.023 |
| 2018/09/10 13:50 | 0.0572 | 0.16 | 0.074 | 0.066 | - |
| 2018/09/11 13:00 | 0.0065 | 0.53 | 0.038 | 0.022 | 0.008 |
| 2018/09/11 13:10 | 0.0078 | 0.50 | 0.038 | 0.023 | 0.004 |
| 2018/09/11 13:20 | 0.0124 | 0.46 | 0.047 | 0.027 | 0.008 |
| 2018/09/11 13:40 | 0.0163 | 0.37 | 0.051 | 0.034 | 0.014 |
| 2018/09/11 13:50 | 0.0228 | 0.33 | 0.054 | 0.036 | 0.010 |
| 2018/09/11 14:00 | 0.0260 | 0.28 | 0.070 | 0.053 | 0.012 |
| 2018/09/11 14:10 | 0.0345 | 0.23 | 0.071 | 0.053 | 0.031 |
| 2018/09/11 14:20 | 0.0449 | 0.18 | 0.070 | 0.060 | 0.037 |
| 2018/09/11 14:30 | 0.0585 | 0.14 | 0.078 | 0.077 | - |

Table R4. Measured flow variables in the model and field mangrove forest by Maza et al. (2017) and Yoshikai et al. (2022), respectively, and the settings of model forcing and target variables to reproduce in the application to the respective mangrove forest.

| | Model mangrove forest in Maza et al. (2017) | Field mangrove forest in Yoshikai et al. (2022) |
| --- | --- | --- |
| Measured flow variables | $h$, $U$, $u(z)$, $k(z)$ | $h$, $\Delta\eta$, $u(z)$, $U$, $\tau_{bed}$ |
| Controlled variables in the model | $h$, $U$ | $h$, $\Delta\eta$ |
| Target variables to reproduce | $u(z)$, $k(z)$ | $u(z)$, $U$, $\tau_{bed}$ |

**Reviewer [1.3]:**

2) The sediment transport computations seem arbitrary from the choice of parameters. Although it is interesting to compare sedimentation rates for the different model parameterizations, the rates need to be validated by data to be able to say that they are realistic. Especially the choice of just one setting seems very limited, depending on the types of grain sizes and parameterization the rates can be very different and there is now no indication that the model can predict realistic rates or that the new model predicts these rates "better". I suggest to add a validation here and test a wide range of sediment parameters to be able to identify trends.

**Response [1.3]:**

As described in Response [1.1], currently the data set that can be used for model validation on sediment transport in mangrove forests is unavailable from both flume- and field-based studies. Thus, we are not able to address the reviewer's suggestion on the model validation of sediment transport in this study. Furthermore, the suggested additional analyses on the wide range of sediment parameters would take the study beyond its original scope, that is, a realization of the realistic flow simulation in *Rhizophora* mangrove forests. Therefore, as suggested in the comment [1.5], we would like to remove the sections on sediment transport simulation in the revised manuscript to underscore our contribution in this study.

**Reviewer [1.4]:**

3) Based on the analyses presented, some parts of the discussion overstate the outcome of the study, for example :

line 434: "The good performance of the model in both the model- and real-Rhizophora mangrove forests having a range of vegetation complexity (Fig. 3) suggests the model's general applicability to Rhizophora mangrove forests worldwide. "- I don't think you can state that the model improves predictions for any other systems than the one studied here. To be able to upscale to other systems, more analyeses are needed.

line 458: "For the practical use of the model, we proposed a model framework (Fig. 1) leveraging an empirical model for the Rhizophora root system (Rhroot model) with parameterization of subgrid-scale tree variations (Fig. 2), which we implemented in COAWST."- as far as I understand you implemented the already existing theoretical model of root area, so more careful phrasing here since the novel part here is the implementation.

line 472: "This study thus offers the first framework of numerical modeling which can be readily applied to Rhizophora mangrove forests in the field." - again, I think you with this work you provide a good implementation of the model

**Response [1.4]:**
Regarding L434, we agree with the reviewer's point. Especially, model application to natural mangrove forests that may have heterogeneous tree sizes and distribution is currently lacking, which would require further studies to confirm the model's general applicability. The model applicability to denser mangrove forests (e.g., forests having $a > 0.9$ m$^{-1}$) may also need confirmation in future studies. Therefore, we would like to revise the sentence as follows:

> "The good performance of the model in both the model- and real-*Rhizophora* mangrove forests suggests the model's applicability to forests having the vegetation density $a$ in the range 0.09–0.9 m$^{-1}$ near the bottom (Fig. 3) and an in-line tree distribution like planted forests. However, the applicability to forests having $a > 0.9$ m$^{-1}$ and/or heterogeneous tree sizes and distribution, a condition often observed in natural mangrove forests, needs further investigation in future studies."

Regarding L458, we would like to revise the sentence as follows:

> "For the practical use of the model, we implemented in COAWST an empirical model for the *Rhizophora* root system (*Rh*-root model; Fig. 1) with parameterization of subgrid-scale tree variations (Fig. 2) that enables the model application without rigorous measurements of root structures."

Regarding L472, we would like to revise the sentence as follows:

> "Therefore, the model presented in this study may realize a realistic forest-scale numerical modeling of flows in *Rhizophora* mangrove forests in the field."

**Reviewer [1.5]:**

4) The discussion on the sediment transport would need to be removed or revised in case new analyses are added

**Response [1.5]:**

We would like to remove the results and discussions on the sediment transport simulation in the revised manuscript as suggested. Please also see our responses [1.1] and [1.3].

**Reviewer [1.6]:**

Minor comments:

1) the paragraph on carbon in the introduction seems a bit far from what is presented in the study

**Response [1.6]:**

We agree with the reviewer. We will remove the said paragraph in the revised manuscript.

**Reviewer [1.7]:**

2) I am not sure you need the reference runs in your graphs since you are comparing the new root structures with static vegetation. Why not add the model in Xie et al (2020) to compare with another "more realistic" representation of roots?

**Response [1.7]:**

We assume that the reviewer is referring to the reference runs to the simulations using the cylinder model shown in Figs. 5 and 7.

We consider that the inclusion of the cylinder model runs is important to show how much the new model could improve the accuracy of flow predictions for *Rhizophora* mangrove forests compared to the conventional drag parameterization using the array of cylinders. Thus, we would like to keep the results and discussions on the cylinder drag model in the manuscript.

As suggested by the reviewer, we have added a simulation case using the model of root system structures used in Xie et al. (2020) in our analysis (denoted as Xie root model hereafter). Below, we describe the Xie root model, its implementation to the COAWST, and some results and discussions. Please note that we have added another simulation case using an increased bed roughness based on the suggestion by Reviewer #2; Please also see Response [2.2].

We examined the use of the root model used in Xie et al. (2020) as a predictor of $a_{root}$ in Eq. (1). In Xie et al. (2020), the shape of roots was simplified to cylindrical objects with a fixed diameter and height, hence to the array of vertical cylinders. The number of roots of a tree is given by a function of stem diameter as:

$$n_{root,ind} = n_{root,max} \frac{1}{1+exp\left[f_{root}\left(\frac{D_{stem,max}}{2}-D_{stem}\right)\times 0.01\right]} \qquad (R1)$$

where $n_{root,ind}$ is the number of roots of a tree having a stem diameter of $D_{stem}$ (m), $n_{root,max}$ is the maximum number of roots of a tree, $f_{root} = 0.1$ is a constant describing the rate of increase, $D_{stem,max}$ is the maximum stem diameter (m), and the factor 0.01 is for the unit conversion of stem diameter from meter to centimeter. In Xie et al. (2020), the parameters are set as $n_{root,max} = 5000$, $D_{stem,max} = 1.0$ (m) for *Rhizophora* trees. In addition, Xie et al. (2020) gave the root diameter ($D_{root}$) and height ($H_{root}$) as $D_{root} = 0.01$ m and $H_{root} = 0.15$ m, respectively.

We applied the Xie root model to the field mangrove setting of Bakhawan Ecopark. We used the measured mean stem diameter $D_{stem,ave} = 0.066$ m (Table 2) for $D_{stem}$ in Eq. (R1), then calculated the $n_{root,ind}$ with the same parameter setting as Xie et al. (2020). The $a_{root}$, which is used for calculating the drag by the roots in Eq. (1), is then given as

$$a_{root} = n_{tree}n_{root,ind}D_{root} \quad \text{for } z \leq H_{root} \tag{R2a}$$

$$a_{root} = 0 \qquad\qquad\qquad \text{for } z > H_{root} \tag{R2b}$$

Furthermore, in the turbulence dissipation terms of Eq. (6a–b), $D_{stem,ave} = 0.066$ m and $D_{root} = 0.01$ m were applied for $L_{stem}$ and $L_{root}$, respectively.

Figure R3, which is a revision of Fig. 3, shows the vegetation projected area density predicted using the Xie root model applied to the field mangrove forest. The Xie root model predicted the vegetation projected area density near the bed as 0.3 m$^{-1}$, which is significantly lower than the measured value. In addition, due to the limited root height ($H_{root} = 0.15$ m), it resulted in the significantly underestimated vegetation projected area density ($a$) throughout the depths.

A comparison of the modeled velocity profiles with measurements is provided in Fig. R4d, which is a revision of Fig. 5. As expected, the use of Xie root model lead significant overestimation of velocities. Although the shape of $a$ predicted by Xie root model resembles those of submerged vegetations, the predicted velocity profiles did not show a prominent velocity inflection between within and above the canopy layer (root zone in this case), a profile typically observed in the flows in a region with submerged vegetations (e.g., King et al., 2012; Nepf, 2012). This may be due to the low vegetation area density predicted by the Xie root model that was not dense enough to generate the velocity inflection. Nepf (2012) suggested that the velocity profile in a region with submerged vegetations exhibits a boundary-layer form with no inflection point if $C_D a h_v < 0.04$ (where $h_v$ is the height of vegetation). Considering the similar factor for the root zone ($C_D a H_{root}$) in our analysis and assuming that $C_D = 1.0$, it is estimated as 0.045, which is very close to the limit generating the boundary layer profile suggested by Nepf (2012).

A comparison with the time-series data is provided in Fig. R5. Similar to the trend seen in Fig. 4d, the use of Xie root model resulted in consistently higher cross-sectional and near-bottom velocities compared to the measured values. Consequently, the bed shear stress was significantly overestimated.

As an action for manuscript revision, we will add the condensed version of the above descriptions on model testing using the Xie root model and its results and discussions in the revised manuscript. We will also replace Fig. 3 with Fig. R3, Fig. 5 with Fig R4, and include Fig. R5 as Fig. 8 in the revised manuscript.

[Figure]

Figure R3. Vertical profiles of vegetation projected area density, $a$, in (a) a model *Rhizophora* mangrove forest examined by Maza et al. (2017) and (b) a real *Rhizophora* mangrove forest examined by Yoshikai et al. (2022), where the values were calculated with $dz$ = 0.05 m vertical interval (markers). $HR_{max}$ is the maximum root height (2.01 m in Maza et al. (2017); Table 1). The modeled $a$ in panel "b" is given by the *Rhizophora* root module using the parameters shown in Tables 2 and S1 (for Bak2). The projected area density of cylinder arrays (in panels "a" and "b") as well as the $a$ predicted using the root model of Xie et al. (2020) (in panel "b"), which were used for comparison with the new model to represent the impacts of *Rhizophora* mangroves, is also shown (dashed lines).

[Figure]

Figure R4. Comparison of the vertical profiles of velocity (*u*) predicted by the COAWST employing (a) *Rh* model using actual and modeled root projected area density profile ($a_{root}$), (b) cylinder model with sparse and (c) dense array, (d) Xie root model, (e) increased bed roughness as an approximation of vegetation drag, and (d) without imposing vegetation drag (no vegetation), and measurement by Yoshikai et al. (2022) for some selected tidal phases during the measurement period. Root mean square error (RMSE) and $R^2$ values of the modeled *u* against the measured data are also shown, for which computation of the predicted value at the height of the measurement point was obtained by the interpolation of *u* computed at adjacent vertical layers.

[Figure]

Figure R5. Time-series of measured and predicted (a, d) cross-sectional mean velocity ($U$), (b, e) (spatially averaged) velocity at $z = 0.05$ m, and (c, f) bed shear stress ($\tau_{bed}$) during the two-days measurement in Bakhawan Ecopark. The measured values are from Yoshikai et al. (2022) and the predicted values are obtained through the COAWST employing the Xie root model and the increased bed roughness as an approximation of drag by mangroves, respectively.

**Reviewer [1.8]:**
line 91: the reference seems to be the data of the paper. I would like to know what is different between the implemented model and the model you are referring to here

**Response [1.8]:**
The reference here (Yoshikai et al., 2021) is the paper presenting an empirical model for the *Rhizophora* root system structures (referred to as *Rh*-root model in this manuscript), which we implemented in the COAWST in this study. Therefore, the model we are referring to here is the same as the one implemented in the COAWST.

**Reviewer [1.9]:**
line 181: are you defining tree sizes as a distribution?

**Response [1.9]:**

We did not impose spatial variations in tree sizes in the model testing performed in this study, as described in L217. However, the model presented in this study has the capability of accounting for the spatially variable tree parameters (by defining variable stem diameter and tree density in each grid) which may be needed in a large-scale, such as a forest-scale, flow simulations in a mangrove forest as discussed in L463–474.

**Reviewer [1.10]:**

table 2: please make more clear that these are the measurements by linking them to the map

**Response [1.10]:**

We will add a sentence in the caption of Table 2 referring to Fig. R1c for where the hydrodynamic and vegetation parameters came from. We will also indicate in the table which parameters were measured in the previous studies and which ones were calibrated or assumed in the model.

**Reviewer [1.11]:**

line 259: please present the sensitivity runs in the supplementary

**Response [1.11]:**

We provided the results of sensitivity runs of varying $\gamma$ in the prediction of the vertical profile of turbulent kinetic energy in Fig. S6. We will include the figure in the Supporting Information in the revised manuscript as suggested.

[Figure]

Figure R6. Root mean square error (RMSE) of modeled turbulent kinetic energy ($k$) against the measured data in (a) Exp 1, (b) Exp2, and (c) both Exp 1 and 2, by varying the value of scale coefficient ($\gamma$), for which the computation of the predicted value at the height of the measurement point was obtained by the interpolation of $k$ computed at adjacent vertical layers.

**Reviewer [1.12]:**

fig 4: maybe remind the reader in the caption what is HRmax

**Response [1.12]:**

We will add a sentence in the caption of Fig. 4 explaining that $HR_{max}$ stands for the maximum root height.

**Reviewer [1.13]:**

Fig 5: what absolute water levels and time-steps throughout the tidal cycle are presented here? I am not sure what the difference is between actual and modeled aroot. Is one the data and one the predictions of the implemented model? Add R^2 values to quantify the error

**Response [1.13]:**

We have provided the information on water levels in Table S2 in the analysis shown in Fig. 5. The model was run with a time-step of 2 seconds. However, please note that although the model was compared with the time-series data of the tidal cycle, we have created a steady state of flow in the model for the comparison with each measured data as described in L222–223 and Response [1.2].

It is correct that the $Rh$ model with actual $a_{root}$ used the measured data of $a_{root}$ while the $Rh$ model with modeled $a_{root}$ used the predicted $a_{root}$ of the implemented $Rh$-root model, as described in L283–284. We have made this point clearer in Table R5; please see Response [2.5] below.

Finally, we have added $R^2$ and RMSE values in the figure. Please see Fig. R4 shown above.

**Reviewer [1.14]:**

I hope that the authors can extent their analyses and revise the manuscript as I believe this will be a very useful contribution.

**Response [1.14]:**

We thank the reviewer again for the constructive comments. We believe that our proposed revision would improve the manuscript substantially.

**References**

King, A. T., Tinoco, R. O., & Cowen, E. A. (2012). A k–ε turbulence model based on the scales of vertical shear and stem wakes valid for emergent and submerged vegetated flows. Journal of Fluid Mechanics, 701, 1–39. https://doi.org/10.1017/jfm.2012.113.

Maza, M., Adler, K., Ramos, D., Garcia, A. M., & Nepf, H. (2017). Velocity and drag evolution from the leading edge of a model mangrove forest. Journal of Geophysical Research: Oceans, 122(11), 9144–9159. https://doi.org/10.1002/2017JC012945.

Nepf, H. M. (2012). Flow and transport in regions with aquatic vegetation. Annual review of fluid mechanics, 44, 123–142. https://doi.org/10.1146/annurev-fluid-120710-101048.

Xie, D., Schwarz, C., Brückner, M. Z., Kleinhans, M. G., Urrego, D. H., Zhou, Z., & Van Maanen, B. (2020). Mangrove diversity loss under sea-level rise triggered by bio-morphodynamic feedbacks and anthropogenic pressures. Environmental Research Letters, 15(11), 114033. https://doi.org/10.1088/1748-9326/abc122.

Yoshikai, M., Nakamura, T., Bautista, D. M., Herrera, E. C., Baloloy, A., Suwa, R., Basina, R., Primavera-Tirol, Y. H., Blanco, A.C., & Nadaoka, K. (2022) Field measurement and prediction of drag in a planted Rhizophora mangrove forest. Journal of Geophysical Research: Oceans, 127, e2021JC018320. https://doi.org/10.1029/2021JC018320.

Yoshikai, M., Nakamura, T., Suwa, R., Argamosa, R., Okamoto, T., Rollon, R., ... & Nadaoka, K. (2021). Scaling relations and substrate conditions controlling the complexity of Rhizophora prop root system. Estuarine, Coastal and Shelf Science, 248, 107014. https://doi.org/10.1016/j.ecss.2020.107014.

**Response to Reviewer #2**

**Reviewer [2.1]:**
**General Comments**

This manuscript presents a new approach to modeling the flow of water within Rhizophora mangroves. The key improvements to the COAWST vegetation package are: (1) allowing the vertical varying projected area density (frontal area per unit plan area), (2) using the root and stem length-scales in the turbulence dissipation terms, (3) implementing the Rhizophora module which can calculate projected area density from easily obtainable field measurements. These improvements allow the field to move beyond the conventional cylinder assumption, and are generally applicable to all hydrodynamically rough environments which aren't well described by cylinders.

I liked the approach and theme of this paper, and think with some revisions it would be a nice contribution. I also really appreciated the detail of the supplemental information.

**Response [2.1]:**
We thank the reviewer for the thorough assessment and constructive comments on our manuscript. Please see our responses to the comments below.

**Reviewer [2.2]:**
**Specific Comments**

1. The no vegetation case shown in Fig. 5 and Fig S1.

   1. While including this test case in the manuscript is interesting to see how the hydrodynamics are changed if the mangroves were removed from the ecosystem, it seems to change direction from the rest of the paper. I believe the message of this paper is comparing how this new approach of accounting for the roughness of mangroves is different from past approaches (cylinder arrays or enhanced z0 values). I think that if the z0 value for the no-vegetation case is increased, maybe similarly to the Zhang 2012 mentioned on line 65, this would fit with the theme of figures 5 and 6 which contrast the new approach to past approaches.

**Response [2.2]:**
We thank the reviewer for the suggestion. Because we think that the results of the no vegetation case shown in Fig. 5 and Fig. S1 are important for demonstrating how much the drag by mangrove forests could be significant in affecting the flows in mangrove forests, hence strengthening the importance of proper parameterization of the impacts of mangroves, we would like to keep them in the manuscript.

We agree with the reviewer that a case with increased bed roughness ($z_0$ value) would fit the theme of the paper and add some insights into the effects of different drag parameterization.

Below, we describe how the bed shear stress is computed in the COAWST, and how the model is tested using the increased $z_0$ value as an approximation of mangrove drag.

In the COAWST, bed shear stress is computed based on quadratic law using the velocities at the bottom computational cell as (Warner et al., 2008):

$$\tau_{bed} = \rho_w C_{bed} u^2 \tag{R3}$$

where $\tau_{bed}$ is the bed shear stress (N m$^{-2}$), $\rho_w$ is the water density (kg m$^{-3}$), $C_{bed}$ is the bed drag coefficient, and $u$ is the flow velocity (m s$^{-1}$) computed at the bottom cell. It assumes that the flow in the bottom boundary layer has the classic vertical logarithmic profile as:

$$|u| = \frac{u_*}{\kappa} \ln\left(\frac{z_{bottom}}{z_0}\right) \tag{R4}$$

where $u_*$ is the friction velocity, $\sqrt{\tau_{bed}}$, $\kappa = 0.41$ is the von Kármán constant, $z_{bottom}$ is the mid-elevation point of the bottom cell above the bed (m), and $z_0$ is the bed roughness length (m). From Eqs. (R3)–(R4), the relationship of $C_{bed}$ and $z_0$ is:

$$C_{bed} = \kappa^2 \left[\ln\left(\frac{z_{bottom}}{z_0}\right)\right]^{-2} \tag{R5}$$

The value of $z_0$ or $C_{bed}$ can be related to the Manning's coefficient ($n_{manning}$) as follows considering turbulent open channel flow. In an open channel flow with depth-averaged velocity $U_{mean}$, water depth $h$, and bed slope $S_0$, the $U_{mean}$ can be described using the Manning's coefficient as

$$U_{mean} = \frac{1}{n_{manning}} h^{2/3} S_0^{1/2} \tag{R6}$$

Assuming the steady flow where the momentum balance can be reduced to an equilibrium between the bed shear stress $\tau_{bed}$ and the gravitational (or pressure) forces driving the flow, the bed shear stress can be expressed as (Crompton et al., 2020):

$$\tau_{bed} = \rho_w g h S_0 \tag{R7}$$

where $g$ is the gravitational acceleration (m s$^{-2}$). From Eq. (R6)–(R7) and assuming that the depth-averaged form of Eq. (R3), $\tau_{bed} = \rho_w C_{bed,mean} U_{mean}^2$, is valid, the Manning's coefficient can be expressed as:

$$n_{manning} = h^{1/6} \sqrt{\frac{C_{bed,mean}}{g}} \tag{R8}$$

where $C_{bed,mean}$ is the bed drag coefficient which is used for computing $\tau_{bed}$ using the $U_{mean}$. Also, by relating the depth-averaged form of Eq. (R5), $C_{bed,mean}$ can be expressed using $z_0$ as (Lenz et al., 2017):

$$C_{bed,mean} = \kappa^2 \left[\ln\left(\frac{h}{z_0}\right)\right]^{-2} \tag{R9}$$

Considering the Manning's coefficient of 0.14, which is a value typically used for approximating the drag by mangroves (e.g., Zhang et al., 2012), and a water depth of 0.5 m, based on Eqs. (R8)–(R9), the equivalent bed roughness $z_0$ to that is 0.22 m.

As suggested by the reviewer, we performed an additional model analysis using an increased $z_0$ value as an approximation of mangrove drag. However, the application of Eqs. (R3)–(R5) needs a condition $z_0 < z_{bottom}$, which limits the applicable $z_0$ value depending on the water depth or thickness of the bottom cell. In order to increase the applicable $z_0$ value in our analysis, where the lowest water depth examined was around 0.15 m (Fig. 6; Table R3), we reduced the number of vertical layers from 5 to 3, which increased the minimum $z_{bottom}$ up to 0.025 m. We then conducted the analysis using $z_0 = 0.02$ m as a case of increased $z_0$; however, this value is considered lower compared to the typical Manning's coefficient value of 0.14 (of which the equivalent value is $z_0 = 0.22$ m under the water depth 0.5 m)

Results of the model analysis with the increased $z_0$ are provided in Figs. R4e and R5, which can be seen below. The model predicted the significant attenuation of flow velocity from the surface to the bottom due to the large bottom friction produced by the increased $z_0$, which did not well represent the actual conditions of the velocity profile in the *Rhizophora* mangrove forest (Fig. R4e).

Comparison with the time-series data showed contrasting results with the one using the Xie root model (please see Response [1.7] for the descriptions of the case using Xie root model). Although both cases showed a large overestimation of flow velocity when the water depth is relatively high (e.g., $h > 0.3$ m), the case using the increased $z_0$ approached the measured data with decreasing water depth while the case using the Xie root model consistently overestimated the velocity throughout the measurement period (Fig. R5a–e). This different trend is due to the different drag parameterization with the bed roughness or objects within the water column. Specifically, the decrease in the total projected area of submerged objects exerting drag with a decrease in water depth cannot be accounted for by the mangrove drag approximation with the bed drag. As a consequence, the bed drag became significant in decelerating the flow velocity when the water depth is low ($h < 0.3$ m) compared to the case using the Xie root model, as seen in the equivalent cross-sectional mean velocity ($U$) to the measured values (Fig. R5a, d). Because bed drag is the main force to counteract the imposed pressure gradient in the increased $z_0$ case, the bed drag showed the overestimation as expected (Fig. R5c, f). This will lead to large overestimation of sediment erosion, suggesting that increased bed roughness does not work well for simulating the sediment transport in mangrove forests.

As an action for manuscript revision, we will add descriptions on the model testing using the increased bed roughness in the Supporting Information and the condensed version of the above descriptions on the results and discussions in the revised manuscript. We will also replace Fig. 5 with Fig R4, and include Fig. R5 as Fig. 8 in the revised manuscript.

[Figure]

Copy of Figure R4 shown above in Response [1.7]: Comparison of the vertical profiles of velocity ($u$) predicted by the COAWST employing (a) *Rh* model using actual and modeled root projected area density profile ($a_{root}$), (b) cylinder model with sparse and (c) dense array, (d) Xie root model, (e) increased bed roughness as an approximation of vegetation drag, and (d) without imposing vegetation drag (no vegetation), and measurement by Yoshikai et al. (2022) for some selected tidal phases during the measurement period. Root mean square error (RMSE) and $R^2$ values of the modeled $u$ against the measured data are also shown, for which computation the predicted value at the height of the measurement point was obtained by the interpolation of $u$ computed at adjacent vertical layers.

[Figure]

Copy of Figure R5 shown above in Response [1.7]: Time-series of measured and predicted (a, d) cross-sectional mean velocity ($U$), (b, e) (spatially averaged) velocity at $z = 0.05$ m, and (c, f) bed shear stress ($\tau_{bed}$) during the two-days measurement in Bakhawan Ecopark. The measured values are from Yoshikai et al. (2022) and the predicted values are obtained through the COAWST employing the Xie root model and the increased bed roughness as an approximation of drag by mangroves, respectively.

**Reviewer [2.3]:**

2. Does the sparse cylinder model (line 289) have an equivalent frontal area to the field data? Similarly to how the cylinder models used in EXP1 and EXP2 have equivalent frontal area to the Rh model. If so, then this section would nicely flow from the lab section where frontal area was conserved. If not, then there is a jump in the methods being used to create the cylinder arrays in the lab vs in the field section. For best transition between the sections I think the method for generating the cylinder arrays should be consistent between the lab and field parts of the paper.

**Response [2.3]:**

The sparse cylinder model applied for the field-based study does not have the equivalent frontal area to the field data in contrast to the cylinder models applied for Exp1 and Exp2 of the laboratory-based study.

We would like to note that the objectives of the model applications to the laboratory- and field-based studies are different. The main objective of the application to the laboratory-based study is to explore the effectiveness of the formulations for the drag and turbulence terms (Eqs. (1)–(6)), which were newly implemented in the COAWST in this study, compared to the cylinder drag model provided the vegetation frontal area density ($a$) as a known parameter. Alternatively, in the case of field studies, the parameter $a$ is usually unknown and needs to be predicted for the model application. Thus, one of the main objectives of the application to the field-based study is to explore the effectiveness of the implemented *Rhizophora* root model – the predictor of $a$ – in combination with the new formulations in the COAWST, compared to the drag parameterizations proposed in previous studies.

The vegetation frontal area density ($a$), which is a labor-intensive parameter to obtain in the field, has remained as a factor making the model application to the field mangrove forests challenging. This is why several modeling studies have parameterized the drag by mangroves in different ways such as cylinder array approximation with arbitrary cylinder density by Xie et al. (2020) (described in L64–66), cylinder array approximation based on the vegetation geometry measured at a height of around 0.25 m by Horstman et al. (2013) (described in L288–289), and increased bed roughness by Zhang et al. (2012) (described in L64–66). In this study, we have examined these drag parameterizations for the application to the field mangrove forest by comparing them with the new model presented in this study, rather than defining the cylinder array having an equivalent frontal area to the field data, assuming that the parameter $a$ is unknown (please note that we have examined the parameterization of Zhang et al. (2012) in the form of dense cylinder arrays – please see L289–291; we have also added new simulation cases using the root model of Xie et al. (2020) and the increased bed roughness – please see Response [1.7] and Response [2.2], respectively).

In the revised manuscript, we will make this point clearer, specifically the difference in the objectives of model applications to the laboratory- and field-based studies.

**Reviewer [2.4]:**

3. I would love to see a figure that shows the difference accounting for 2 length-scales in the turbulence routines makes. This is mentioned on lines 444-445 and lines 424-426. I haven't seen a figure that does this yet, and the changes to the code are already made so I think this could be a nice addition to the paper.

**Response [2.4]:**

As suggested by the reviewer, we have examined the effects of accounting for the different length-scales of stem- and root-generated wakes in the presented model ($Rh$ model). We have done it by performing additional model analyses for the flume experiments that set the two length-scales to either the root diameter ($D_{root,ave}$) or the stem diameter ($D_{stem,ave}$).

The results clearly showed that without accounting for the two different length-scales, the model fails to predict the vertical profile of turbulent kinetic energy ($k$) (Fig. R7). If the two length-scales are set to the stem diameter, the model largely overestimates $k$ specifically in the root zone ($z/HR_{max} < 1$); whereas the model with the length-scales set to the root diameter underestimated $k$ in the upper and above the root zone ($z/HR_{max} > 0.5$). The model that has the length-scales set to the root diameter showed a slight increase in $k$ at the height around $z/HR_{max} = 0.9$ (Fig. R7b), possibly due to the velocity shear generated by the sharp decrease

in frontal area at the top of the root zone (Maza et al., 2017; Fig. 3a). However, this increase in $k$ alone is not enough to explain the significantly higher $k$ at upper and above the root zone compared to the lower root zone, suggesting the significant roles of the different length-scales of stem- and root-generated wakes in shaping the overall structure of $k$.

As an action for manuscript revision, we will include Fig. R7 in the Supporting Information, and a brief discussion on the interpretation of the result described above in the main text.

[Figure]

Figure R7. Comparison of the vertical profiles of (temporally and spatially averaged) velocity ($u$) and turbulent kinetic energy ($k$) normalized by the cross-sectional mean velocity ($U$) predicted by the COAWST using the $Rh$ model with different length-scales of stem- and root-generated wakes ($L_{stem}$ and $L_{root}$, respectively) defined – blue markers: $L_{stem}$ and $L_{root}$ set to the stem diameter ($D_{stem,ave}$) and root diameter ($D_{root,ave}$), respectively; dark-gray markers: $L_{stem}$ and $L_{root}$ both set to $D_{root,ave}$; light-gray markers: $L_{stem}$ and $L_{root}$ both set to $D_{stem,ave}$. The scale coefficient ($\gamma$) was set to 1.2 for all the cases.

**Reviewer [2.5]:**
4. I would appreciate if the cylinder metrics (diameter, density, height) mentioned in EXP1, EXP2, the sparse cylinder array and the dense cylinder array could be compiled into a table and attached as supplemental information.
    1. I believe it is important to state the height of these cylinder arrays. Based on the velocity profiles shown in Figures 4 and 5, I believe all the cylinder arrays span the entire water column. However there are other papers which use cylinder arrays which span a fraction of the water column, so I think it is important to state.

2. The widths are also important to state because of they are used in the turbulence dissipation term. The diameters for EXP1 and EXP2 are already stated, but I couldn't find diameters for the sparce and dense cylinder arrays.

**Response [2.5]:**

We have compiled the parameter settings of all the simulation cases in Table R5. Because we consider that Table R5 is convenient for the readers to grasp the different model configurations used in this study, we would like to include it in the main text of the revised manuscript.

Regarding comment 1, the height of the cylinder arrays was defined well higher than the water level in the model, thus they span the entire water column as pointed out by the reviewer. In the revised manuscript, we will add an explanation of this point. Also, we added a description of the cylinder height in the caption of Table R5.

Regarding comment 2, we noticed that we missed adding the information on the cylinder diameter defined for the field-based study in the manuscript. We have provided the information on the cylinder diameter in Table R5, which will be included in the revised manuscript.

Table R5. Tested model configurations to represent the impact of *Rhizophora* mangroves against flume experiments (Exp1 and 2) in Maza et al. (2017) and field measurement in Yoshikai et al. (2022). In the cylinder model configurations, cylinder height was set well higher than the water level to create the condition that cylinders span the entire water column. $n_{tree}$: tree density; $n_v$: cylinder density; $D_{stem,ave}$: mean stem diameter; $b_v$: cylinder density; $a_{root}$: root projected area density; $D_{root,ave}$: mean root diameter; $z_0$: bed roughness length; $N_{layer}$: number of vertical layers of model grid.

| Test case | Model configuration | Parameter settings | | | | | |
|---|---|---|---|---|---|---|---|
| | | $n_{tree}$ or $n_v$ (m$^{-2}$) | $D_{stem,ave}$ or $b_v$ (m) | $a_{root}$ (m$^{-1}$) | $D_{root,ave}$ (m) | $z_0$ (m) | $N_{layer}$ |
| Flume experiment | *Rh* model | 0.072 ($n_{tree}$) | 0.2 ($D_{stem,ave}$) | Measured value[a] | 0.038 | 0.5 × 10$^{-3}$ | 15 |
| | Cylinder model for Exp1 | 1.22 ($n_v$) | 0.038 ($b_v$) | - | - | 0.5 × 10$^{-3}$ | 15 |
| | Cylinder model for Exp2 | 1.76 ($n_{tree}$) | 0.038 ($D_{stem,ave}$) | - | - | 0.5 × 10$^{-3}$ | 15 |
| Field measurement | *Rh* model with actual $a_{root}$ | 0.36 ($n_{tree}$) | 0.066 ($D_{stem,ave}$) | Measured value[b] | 0.030 | 0.5 × 10$^{-3}$ | 5 |
| | *Rh* model with modeled $a_{root}$ | 0.36 ($n_{tree}$) | 0.066 ($D_{stem,ave}$) | Modeled value[c] | 0.030 | 0.5 × 10$^{-3}$ | 5 |
| | Cylinder model (sparse) | 13.5 ($n_v$) | 0.030 ($b_v$) | - | - | 0.5 × 10$^{-3}$ | 5 |
| | Cylinder model (dense) | 32.3 ($n_v$) | 0.030 ($b_v$) | - | - | 0.5 × 10$^{-3}$ | 5 |
| | Xie et al. root model | 0.36 ($n_{tree}$) | 0.066 ($D_{stem,ave}$) | Eq. (R2)[d] | 0.010 | 0.5 × 10$^{-3}$ | 5 |
| | Increased $z_0$ | - | - | - | - | 0.02 | 3 |
| | No vegetation | - | - | - | - | 0.5 × 10$^{-3}$ | 5 |

[a] Corresponds to the value of black markers minus $n_{tree}D_{stem,ave}$ in Fig. 3a.
[b] Corresponds to the value of black markers minus $n_{tree}D_{stem,ave}$ in Fig. 3b.
[c] Corresponds to the value of blue markers minus $n_{tree}D_{stem,ave}$ in Fig. 3b.

$^d$ Corresponds to the value of light green markers minus $n_{tree}D_{stem,ave}$ in Fig. 3b.

**Reviewer [2.6]:**

5. Lines 200-211, I think that this section and table 1 can be removed from the paper or moved to supplemental information. This section might be useful as a guide for someone using your code, but I don't think it adds much value towards understanding the content of the manuscript.

**Response [2.6]:**

We agree with the reviewer. We would like to move the said section and Table 1 to the Supporting Information in the revised manuscript.

**Reviewer [2.7]:**

6. Generally it seems the core modifications to the code are in the drag term and the turbulence routines, and these change the flow field which then might change the sedimentation rates. I think the sediment parts of this paper are a case study of how these model changes can affect a variable of interest (like sedimentation rates), but I don't think the focus of the paper should be on the changes in sedimentation rates.

**Response [2.7]:**

We agree with the reviewer that the simulations of the sediment transport should not be the focus of the paper. To underscore the contribution of this manuscript, we would like to remove the section in the results and discussion on the sediment transport simulation in the revised manuscript. Please also see Response [1.1], [1.3], and [1.5] inn relation to this point.

**Reviewer [2.8]:**

**Technical corrections/ Typing errors**

1. Line 308 "run" should be "ran"

2. Line 214, could you please consider stating the vertical resolution and/or the number of sigma levels used? This is mentioned in line 241, but I think it would be nice to have all of the domain characteristics mentioned in the same place.

**Response [2.8]:**

We will revise the said point on L308. We will also put information on the vertical layering of the model in the section. We have also included the information on the number of vertical layers of the computational grid in Table R5, which will be included in the revised manuscript.

**References**

Crompton, O., Katul, G. G., & Thompson, S. (2020). Resistance formulations in shallow overland flow along a hillslope covered with patchy vegetation. Water Resources Research, 56(5), e2020WR027194. https://doi.org/10.1029/2020WR027194.

Horstman, E., Dohmen-Janssen, M., & Hulscher, S. J. M. H. (2013, June). Modeling tidal dynamics in a mangrove creek catchment in Delft3D. In Coastal dynamics (Vol. 2013, pp. 833–844).

Lentz, S. J., Davis, K. A., Churchill, J. H., & DeCarlo, T. M. (2017). Coral reef drag coefficients–water depth dependence. Journal of Physical Oceanography, 47(5), 1061-1075.

Maza, M., Adler, K., Ramos, D., Garcia, A. M., & Nepf, H. (2017). Velocity and drag evolution from the leading edge of a model mangrove forest. Journal of Geophysical Research: Oceans, 122(11), 9144–9159. https://doi.org/10.1002/2017JC012945.

Warner, J. C., Sherwood, C. R., Signell, R. P., Harris, C. K., & Arango, H. G. (2008). Development of a three-dimensional, regional, coupled wave, current, and sediment-transport model. Computers & geosciences, 34(10), 1284–1306. https://doi.org/10.1016/j.cageo.2008.02.012.

Xie, D., Schwarz, C., Brückner, M. Z., Kleinhans, M. G., Urrego, D. H., Zhou, Z., & Van Maanen, B. (2020). Mangrove diversity loss under sea-level rise triggered by bio-morphodynamic feedbacks and anthropogenic pressures. Environmental Research Letters, 15(11), 114033. https://doi.org/10.1088/1748-9326/abc122.

Yoshikai, M., Nakamura, T., Bautista, D. M., Herrera, E. C., Baloloy, A., Suwa, R., Basina, R., Primavera-Tirol, Y. H., Blanco, A.C., & Nadaoka, K. (2022) Field measurement and prediction of drag in a planted Rhizophora mangrove forest. Journal of Geophysical Research: Oceans, 127, e2021JC018320. https://doi.org/10.1029/2021JC018320.

Zhang, K., Liu, H., Li, Y., Xu, H., Shen, J., Rhome, J., & Smith III, T. J. (2012). The role of mangroves in attenuating storm surges. Estuarine, Coastal and Shelf Science, 102, 11–23. https://doi.org/10.1016/j.ecss.2012.02.021.

---

## Author Response (AR1)

Dear editor and reviewers,

We thank the reviewers for taking the time to review our manuscript and providing constructive comments. We apologize for the delay in submitting the revised manuscript.

Below, we address the reviewers' comments point by point. In the response, black- and blue-colored characters denote reviewers' comments and our responses, respectively.

**Response to Reviewer #1**

**Reviewer [1.1]:**
Yoshikai et al present a novel implementation of a mangrove root model in the ROMS data package that has the potential to be very useful for the wider scientific community. They show that the model predicts flow velocities and turbulent kinetic energy in a more refined way in the vertical and that this matches qualitatively better with the presented data from a flume experiment and the field. Although I think that the model is very useful and can advance the scientific field, I have major concerns with the application of the model to the sediment transport predictions that links directly to large parts of the discussion. In addition, I missed more detail on the data that has been used. Finally, to help lift this contribution from a presentation of the model, I suggest to apply the model to a case study with available sediment transport rates and potentially morphodynamic change to show that those predictions are improved using the presented model. I outline my concerns below:

**Response [1.1]:**
We thank the reviewer for the thorough assessment and constructive comments on our manuscript.

Regarding the reviewer's concern about the model application to sediment transport, we admit that the model efficacy on sediment transport is not well demonstrated in this study compared to the flow structures in the mangrove forests. While the reviewer's suggestion is interesting and an important topic to address, we would like to note that model application to sediment transport in mangrove forests and its evaluation is currently constrained by data availability. Even for the prediction of flows, comprehensive data sets such as hydrodynamics (e.g., water depth and flow velocity), vegetation parameters (both stems and roots), as well as boundary conditions of the flow (water level gradient in this study) are required to drive and evaluate the developed model that are rarely available especially in the field. The application and evaluation for the sediment transport require additional data such as sediment properties (e.g., grain size), suspended sediment concentration, sedimentation rate, and morphodynamics. To our knowledge, the data set that satisfies such requirement is currently unavailable from both flume- and field-based studies. Hence, at this moment, we are not able to address the reviewer's suggestion regarding the application of the model to a

case study with available sediment transport rates and morphodynamic change to demonstrate the efficacy of the presented model.

Our intention of the sediment transport simulation performed in this study was to demonstrate how the flow field created by the *Rhizophora* mangroves may change the sedimentation rates using the new model. However, given the current limitation to show the model's efficacy on sediment transport and that the main contribution of the study is the realization of the realistic flow simulation in *Rhizophora* mangrove forests, we have removed the results and discussion on the sediment transport simulation in the revised manuscript, as suggested in the comment [1.5]. This is also suggested by Reviewer #2 (please see the comment [2.7]). We believe that the removal of the results on sediment transport simulations made the manuscript more focused on the novelty and the contribution of this study.

We list the changes made below which are related to this point. Please see the marked-up revised manuscript for the detailed changes.

- Title: We removed "sediment transport" from the title, then the title became "Representing the impact of *Rhizophora* mangroves on flow in a hydrodynamic model (COAWST_rh v1.0): the importance of three-dimensional root system structures".
- Abstract: Descriptions on the sediment transport simulations were removed. Accordingly, some sentences related to the aim of the study, overall implications from the results were revised/added.
- Section 2.1 "Model description": Descriptions related to the sediment transport modeling were removed.
- Section 2.3 "Sediment transport simulation": Removed.
- Section 3.3 "Sediment transport simulation": Removed.
- Section 4.2 "Implication for sediment transport in *Rhizophora* mangrove forest": Removed.
- Section 4.3 "Further model improvement": Some discussions previously written in Section 4.2 on the model application to sediment transport simulation were moved to Section 4.3 for discussing the need for further model improvement (L. 620–636 in marked-up version). The first paragraph in the original manuscript (L. 637–641 in marked-up version) was removed because it was referring to the results of sediment transport simulation shown in the original manuscript.
- Conclusion: Descriptions on the sediment transport simulations were removed. Accordingly, some sentences related to the aim of the study, overall implications from the results were revised/added.
- Figure 1: The figure was revised to remove the model linkage to the sediment transport model.
- Figure 8: Removed.
- Table S2: Removed.

**Reviewer [1.2]:**

1) To understand where the data comes from I suggest to add maps of both the study area and the model grid with flow velocities as well as the set-up of the flume experiments/model iof the flume experiments. It is unclear where the measurements have been taken (unvegetated vs. within the root system, close to tidal channels etc) and how exactly the model looks to allow to understand the results and to reproduce the study. In addition the data used for validation should be presented in the supplementary.

**Response [1.2]:**

We have provided in Fig. R1 of this document the maps of the field measurement of Yoshikai et al. (2022) used for the model application in this study, and in Fig. R2 the schematic of the model grid used for testing the model against the flume experiment and field measurement. The set-up of the flume experiment has been described in detail in Maza et al. (2017). Because the reuse of the figures in Maza et al. (2017) in this manuscript would cost us a substantial amount of payment to the publisher of the original article, we would like to refer the reviewers and readers to their original article. We have also provided in Tables R1–3 below the data used for model validation. Furthermore, for a better grasp of the measurements of Maza et al. (2017) and Yoshikai et al. (2022) and the model setting, we have provided Table R4 summarizing the measured flow variables, the setting of model forcing, and the target variables to reproduce for each application.

To avoid any confusion regarding the model setting, we would like to note that the model was tested with the model grid assuming a schematized mangrove forest (Fig. R2) as described in L. 216–225 in the original manuscript, not with a grid representing the actual geometric/topographic conditions of the flume/field. Also, bed elevation and vegetation parameters in the grid were set uniformly over the model domain as described in L. 217–218 in the original manuscript. This simplification of the model setting is deemed reasonable given the (approximately, in the case of the field mangrove forest) spatially uniform vegetation distribution and the well-developed flow conditions at the flow measurement location in both Maza et al. (2017) and Yoshikai et al. (2022) where the dependence of flow structures on the proximity to the forest leading edge is diminished. The flow in the model was driven by a water level gradient imposed between the open boundaries (Fig. R2; L. 218–220 in the original manuscript). We then created a steady state of flow in the model and compared the simulated flow at the monitoring point in the model domain (Fig. R2) with the data (L. 222–223 in the original manuscript). This means that for the model application to the field mangrove forest, the actual time-series of the flow has not been reproduced; rather, steady states of flow were created for each flow measurement.

In the revised manuscript, we included Figs. R1–2 and Tables R1–4 in the Supporting Information as Figs. S1–2 and Tables S3–6, respectively. We also included missing descriptions on the model settings explained above in Section 2.2 in the revised manuscript; please see L. 225–233 and L. 242–244 in the marked-up version.

[Figure]

Figure R1. (a) Satellite image (Google Earth) of the study site of Yoshikai et al. (2022) – Bakhawan Ecopark (red box indicates the area of panel "b"), (b) locations of transect A–B across which the water level gradient was measured together with the hydrodynamic parameters around the reference tree, (c) top view of LiDAR point clouds around the reference tree with information on the locations of trees whose morphological structures were measured, where velocity profiling was conducted, and where sensors were deployed (velocity sensor: electromagnetic velocity meter deployed near the bottom; ADV: Acoustic Doppler Velocimeter deployed to estimate the bed shear stress). It has been shown in Yoshikai et al. (2022) that the average of the velocity measured at the four locations represents well the spatially-averaged values. The point clouds shown were cropped at heights between 0.1–1.7 m for better visualization of the root systems. Figures are modified from Yoshikai et al. (2022).

[Figure]

Figure R2. Model grid (40 × 40 with 5 m horizontal resolution) used for testing the model against laboratory-based and field-based studies. The red circle indicates the location of the monitoring point at which the simulated flow variables were compared with the measured data.

Table R1. Data from the flume experiments of Maza et al. (2017) that were used for the model validation in Figure 4. The values of geometric and flow parameters were converted from the scale in the flume to the real scale. The velocity ($u$) and turbulent kinetic energy ($k$) were taken by averaging the measurements at five lateral positions (ADV3p1–p5; see Fig. 5 of Maza et al., 2017) in the model mangrove forest where the flows were fully developed, which were taken as spatially-averaged values in the mangrove forest. $HR_{max}$: maximum root height, $h$: water depth, $U$: cross-sectional mean velocity, $z$: height above the bed.

| Experiment # | $HR_{max}$ (m) | $h$ (m) | $U$ (m s$^{-1}$) | $z$ (m) | $u/U$ | $k/U^2$ |
|---|---|---|---|---|---|---|
| Exp 1 | 2.016 | 3.0 | 0.31 | 0.08 | 0.54 | 0.012 |
| | | | | 0.32 | 0.62 | 0.013 |
| | | | | 0.56 | 0.66 | 0.015 |
| | | | | 0.80 | 0.64 | 0.032 |
| | | | | 1.04 | 0.69 | 0.026 |
| | | | | 1.28 | 0.75 | 0.024 |
| | | | | 1.52 | 0.84 | 0.053 |
| | | | | 1.76 | 0.97 | 0.035 |
| | | | | 2.00 | 1.05 | 0.033 |
| | | | | 2.24 | 1.10 | 0.043 |
| Exp 2 | 2.016 | 1.79 | 0.58 | 0.08 | 0.75 | 0.018 |

| | | | | | | |
|---|---|---|---|---|---|---|
| | | | | 0.20 | 0.77 | 0.021 |
| | | | | 0.32 | 0.80 | 0.017 |
| | | | | 0.44 | 0.84 | 0.016 |
| | | | | 0.56 | 0.83 | 0.021 |
| | | | | 0.68 | 0.83 | 0.026 |
| | | | | 0.80 | 0.86 | 0.023 |
| | | | | 0.92 | 0.85 | 0.023 |

Table R2. Data from field measurements of Yoshikai et al. (2022) that were used for the model validation in Figure 5. Velocity ($u$) was obtained by averaging the measurements at four locations around the reference tree shown in Fig. R1c which was taken as spatially-averaged values in the mangrove forest.

| Local time | $h$ (m) | $z$ (m) | $u$ (m s$^{-1}$) |
|---|---|---|---|
| 2018/9/10 12:50 | 0.45 | 0.35 | 0.060 |
| | | 0.30 | 0.064 |
| | | 0.25 | 0.060 |
| | | 0.20 | 0.057 |
| | | 0.15 | 0.055 |
| | | 0.10 | 0.044 |
| | | 0.05 | 0.036 |
| 2018/9/10 13:40 | 0.21 | 0.18 | 0.096 |
| | | 0.11 | 0.082 |
| | | 0.04 | 0.059 |
| 2018/9/11 13:00 | 0.53 | 0.45 | 0.046 |
| | | 0.40 | 0.039 |
| | | 0.35 | 0.045 |
| | | 0.30 | 0.044 |
| | | 0.25 | 0.044 |
| | | 0.20 | 0.041 |
| | | 0.15 | 0.034 |
| | | 0.10 | 0.028 |
| | | 0.05 | 0.022 |
| 2018/9/11 14:00 | 0.28 | 0.23 | 0.085 |
| | | 0.14 | 0.072 |
| | | 0.05 | 0.052 |

Table R3. Data from field measurements of Yoshikai et al. (2022) that were used for the model forcing and validation in Figures 6–7 in the original manuscript. The $\Delta\eta$ is the water level difference imposed across the open boundaries in the model (see Fig. R2), $h$ is the water depth, $U$ is the cross-sectional mean flow velocity, $u_{bottom}$ is the spatially-averaged velocity at $z = 0.05$ m, and $\tau_{bed}$ is the bed shear stress.

| Local time | $\Delta\eta$ (m) | $h$ (m) | $U$ (m s$^{-1}$) | $u_{bottom}$ (m s$^{-1}$) | $\tau_{bed}$ (N m$^{-2}$) |
|---|---|---|---|---|---|
| 2018/09/10 12:50 | 0.0143 | 0.45 | 0.050 | 0.036 | 0.023 |
| 2018/09/10 13:10 | 0.0189 | 0.36 | 0.063 | 0.036 | 0.039 |

| | | | | | |
|---|---|---|---|---|---|
| 2018/09/10 13:20 | 0.0273 | 0.32 | 0.064 | 0.041 | 0.032 |
| 2018/09/10 13:40 | 0.0462 | 0.21 | 0.079 | 0.064 | 0.023 |
| 2018/09/10 13:50 | 0.0572 | 0.16 | 0.074 | 0.066 | - |
| 2018/09/11 13:00 | 0.0065 | 0.53 | 0.038 | 0.022 | 0.008 |
| 2018/09/11 13:10 | 0.0078 | 0.50 | 0.038 | 0.023 | 0.004 |
| 2018/09/11 13:20 | 0.0124 | 0.46 | 0.047 | 0.027 | 0.008 |
| 2018/09/11 13:40 | 0.0163 | 0.37 | 0.051 | 0.034 | 0.014 |
| 2018/09/11 13:50 | 0.0228 | 0.33 | 0.054 | 0.036 | 0.010 |
| 2018/09/11 14:00 | 0.0260 | 0.28 | 0.070 | 0.053 | 0.012 |
| 2018/09/11 14:10 | 0.0345 | 0.23 | 0.071 | 0.053 | 0.031 |
| 2018/09/11 14:20 | 0.0449 | 0.18 | 0.070 | 0.060 | 0.037 |
| 2018/09/11 14:30 | 0.0585 | 0.14 | 0.078 | 0.077 | - |

Table R4. Measured flow variables in the model and field mangrove forest by Maza et al. (2017) and Yoshikai et al. (2022), respectively, the variables controlled in the model, and target variables to reproduce for application to the respective mangrove forest.

| | Model mangrove forest in Maza et al. (2017) | Field mangrove forest in Yoshikai et al. (2022) |
|---|---|---|
| Measured flow variables | $h$, $U$, $u(z)$, $k(z)$ | $h$, $\Delta\eta$, $u(z)$, $U$, $\tau_{bed}$ |
| Controlled variables in the model | $h$, $U$ | $h$, $\Delta\eta$ |
| Target variables to reproduce | $u(z)$, $k(z)$ | $u(z)$, $U$, $\tau_{bed}$ |

**Reviewer [1.3]:**

2) The sediment transport computations seem arbitrary from the choice of parameters. Although it is interesting to compare sedimentation rates for the different model parameterizations, the rates need to be validated by data to be able to say that they are realistic. Especially the choice of just one setting seems very limited, depending on the types of grain sizes and parameterization the rates can be very different and there is now no indication that the model can predict realistic rates or that the new model predicts these rates "better". I suggest to add a validation here and test a wide range of sediment parameters to be able to identify trends.

**Response [1.3]:**

As described in Response [1.1], currently the data set that can be used for model validation on sediment transport in mangrove forests is unavailable from both flume- and field-based studies. Thus, we are not able to address the reviewer's suggestion on the model validation of sediment transport in this study. Furthermore, the suggested additional analyses on the wide range of sediment parameters would take the study beyond its original scope, that is, a realization of the realistic flow simulation in *Rhizophora* mangrove forests. Therefore, as suggested in the comment [1.5], we removed the sections on sediment transport simulation in the revised manuscript to underscore our contribution in this study. Please see Response [1.1] for the changes made related this point.

**Reviewer [1.4]:**

3) Based on the analyses presented, some parts of the discussion overstate the outcome of the study, for example :

line 434: "The good performance of the model in both the model- and real-Rhizophora mangrove forests having a range of vegetation complexity (Fig. 3) suggests the model's general applicability to Rhizophora mangrove forests worldwide. "- I don't think you can state that the model improves predictions for any other systems than the one studied here. To be able to upscale to other systems, more analyeses are needed.

line 458: "For the practical use of the model, we proposed a model framework (Fig. 1) leveraging an empirical model for the Rhizophora root system (Rhroot model) with parameterization of subgrid-scale tree variations (Fig. 2), which we implemented in COAWST."- as far as I understand you implemented the already existing theoretical model of root area, so more careful phrasing here since the novel part here is the implementation.

line 472: "This study thus offers the first framework of numerical modeling which can be readily applied to Rhizophora mangrove forests in the field." - again, I think you with this work you provide a good implementation of the model

**Response [1.4]:**

Regarding L. 434, we agree with the reviewer's point. Especially, model application to natural mangrove forests that may have heterogeneous tree sizes and distribution is currently lacking, which would require further studies to confirm the model's general applicability. The model applicability to denser mangrove forests (e.g., forests having $a > 0.9$ m$^{-1}$) may also need confirmation in future studies. Therefore, we revised the sentence as follows:

> L. 528–513 in the marked-up version: "The good performance of the model in both the model- and real-*Rhizophora* mangrove forests suggests the model's applicability to forests having the vegetation density $a$ in the range 0.09–0.9 m$^{-1}$ near the bed (Fig. 3) and an in-line tree distribution like planted mangrove forests. However, the applicability to forests having $a > 0.9$ m$^{-1}$ and/or heterogeneous tree sizes and distribution, a condition often observed in natural mangrove forests, needs further investigation in future studies."

Regarding L. 458, we revised the sentence as follows:

> L. 556–569 in the marked-up version: "For the practical use of the model, we implemented in COAWST an empirical model for the *Rhizophora* root system (*Rh*-root model; Fig. 1) with parameterization of subgrid-scale tree variations (Fig. 2) that enables the model application without rigorous measurements of root structures."

Regarding L. 472, we revised the sentence as follows:

L. 570–571 in the marked-up version: "Therefore, the model presented in this study may realize a realistic forest-scale numerical modeling of flows in *Rhizophora* mangrove forests in the field."

**Reviewer [1.5]:**

4) The discussion on the sediment transport would need to be removed or revised in case new analyses are added

**Response [1.5]:**

We have removed the results and discussions on the sediment transport simulation in the revised manuscript as suggested. Please also see our responses [1.1] and [1.3].

**Reviewer [1.6]:**

Minor comments:

1) the paragraph on carbon in the introduction seems a bit far from what is presented in the study

**Response [1.6]:**

We agree with the reviewer. We have removed the said paragraph in the revised manuscript (L. 49–57 in the marked-up version).

**Reviewer [1.7]:**

2) I am not sure you need the reference runs in your graphs since you are comparing the new root structures with static vegetation. Why not add the model in Xie et al (2020) to compare with another "more realistic" representation of roots?

**Response [1.7]:**

We assume that the reviewer is referring to the reference runs to the simulations using the cylinder model shown in Figs. 5 and 7 in the original manuscript.

We consider that the inclusion of the cylinder model runs is important to show how much the new model could improve the accuracy of flow predictions for *Rhizophora* mangrove forests compared to the conventional drag parameterization using the array of cylinders. Thus, we would like to keep the results and discussions on the cylinder drag model in the manuscript.

As suggested by the reviewer, we have added a simulation case using the model of root system structures used in Xie et al. (2020) in our analysis (denoted as Xie root model hereafter). Below, we describe the Xie root model, its implementation to the COAWST, and some results and discussions. Please note that we have added another simulation case using an increased bed roughness based on the suggestion by Reviewer #2; Please also see Response [2.2].

We examined the use of the root model used in Xie et al. (2020) as a predictor of $a_{root}$ in Eq. (1). In Xie et al. (2020), the shape of roots was simplified to cylindrical objects with a fixed diameter and height, hence to the array of vertical cylinders. The number of roots of a tree is given by the function of stem diameter as:

$$n_{root,ind} = n_{root,max} \frac{1}{1+exp\left[f_{root}\left(\frac{D_{stem,max}}{2}-D_{stem}\right)\times 100\right]} \qquad (R1)$$

where $n_{root,ind}$ is the number of roots of a tree having a stem diameter of $D_{stem}$ (m), $n_{root,max}$ is the maximum number of roots of a tree, $f_{root}$ = 0.1 is a constant describing the rate of increase of roots with $D_{stem}$, $D_{stem,max}$ is the maximum stem diameter (m), and the factor 100 is for the unit conversion of stem diameter from meter to centimeter. In Xie et al. (2020), the parameters are set as $n_{root,max}$ = 5000, $D_{stem,max}$ = 1.0 (m) for *Rhizophora* trees. In addition, Xie et al. (2020) gave the root diameter ($D_{root}$) and height ($H_{root}$) values as $D_{root}$ = 0.01 m and $H_{root}$ = 0.15 m, respectively.

We applied the Xie root model to the field mangrove setting of Bakhawan Ecopark. We used the measured mean stem diameter $D_{stem,ave}$ = 0.066 m (Table 2 in the original manuscript) for $D_{stem}$ in Eq. (R1), then calculated the $n_{root,ind}$ with the same parameter setting as Xie et al. (2020). The $a_{root}$, which is used for calculating the drag by the roots in Eq. (1), is then given as:

$$a_{root} = n_{tree}n_{root,ind}D_{root} \quad \text{for } z \leq H_{root} \qquad (R2a)$$

$$a_{root} = 0 \qquad\qquad\qquad \text{for } z > H_{root} \qquad (R2b)$$

In addition, in the turbulence dissipation terms of Eq. (6a–b), $D_{stem,ave}$ = 0.066 m and $D_{root}$ = 0.01 m were applied for $L_{stem}$ and $L_{root}$, respectively.

Figure R3, which is a revision of Fig. 3 in the original manuscript, shows the vegetation projected area density predicted using the Xie root model applied to the field mangrove forest. The Xie root model predicted the vegetation projected area density near the bed as 0.3 m$^{-1}$, which is significantly lower than the measured value. In addition, due to the limited root height ($H_{root}$ = 0.15 m), it resulted in the significantly underestimated vegetation projected area density ($a$) throughout the depths.

A comparison of the modeled velocity profiles with measurements is provided in Fig. R4d, which is a revision of Fig. 5. As expected, the use of Xie root model lead significant overestimation of velocities. Although the shape of $a$ predicted by Xie root model resembles those of submerged vegetations, the predicted velocity profiles did not show a prominent velocity inflection between within and above the canopy layer (root zone in this case), a profile typically observed in the flows in a region with submerged vegetations (e.g., King et al., 2012; Nepf, 2012). This may be due to the low vegetation area density predicted by the Xie root model that was not dense enough to generate the velocity inflection. Nepf (2012) suggested that the velocity profile in a region with submerged vegetations exhibits a boundary-layer form with no inflection point if $C_{D}ah_v$ < 0.04 (where $h_v$ is the height of vegetation). Considering the similar factor for the root zone ($C_{D}aH_{root}$) in our analysis and

assuming that $C_D$ = 1.0, it is estimated as 0.045, which is very close to the limit generating the boundary layer profile suggested by Nepf (2012).

A comparison with the time-series data is provided in Fig. R5. Similar to the trend seen in Fig. 4d, the use of Xie root model resulted in consistently higher cross-sectional and near-bottom velocities compared to the measured values. Consequently, the bed shear stress was significantly overestimated.

In the revised manuscript, we have included the condensed version of the above descriptions as followings:

- Text S6 in the Supporting Information: The description of the Xie root model
- L. 331–332, 339–343 in the marked-up version (Section 2.2.2): The description on the use of the Xie root model for model testing
- L. 406–408, 437–439 in the marked-up version (Section 3.2): The description on the results of the use of Xie root model
- L. 497–505 in the marked-up version (Section 4.1): The discussions on the results of the use of Xie root model
- Figure 3b: Replaced with Fig. R3b.
- Figure 5: Replaced with Fig. R4.
- Figure 8: Replaced with Fig. R5

Additionally, given the increase in the discussions on the model performance, Section 4.1 "Model performance" in the original manuscript was split into two sections Section 4.1 "Performance of the previously proposed drag parameterization" and Section 4.2 "Performance of the new model".

Finally, as we have updated the model code to include the option to use the Xie root model, and created new files for running the new model test cases, we have issued a new DOI for the updated model code and data archived in Zenodo. The "Code and data availability" section (L. 672 in the marked-up version) was revised accordingly.

[Figure]

Figure R3. Vertical profiles of vegetation projected area density, *a*, in (a) a model *Rhizophora* mangrove forest examined by Maza et al. (2017) and (b) a real *Rhizophora* mangrove forest examined by Yoshikai et al. (2022), where the values were calculated with *dz* = 0.05 m vertical interval (markers). $HR_{max}$ is the maximum root height (2.01 m in Maza et al. (2017); Table 1). The modeled *a* using the *Rh*-root model in panel "b" is given by the *Rhizophora* root module using the parameters shown in Tables 2 and S1 (for Bak2). The projected area density of cylinder arrays (in panels "a" and "b") as well as the *a* predicted using the root model of Xie et al. (2020) (in panel "b"), which were used for comparison with the new model to represent the impacts of *Rhizophora* mangroves, is also shown (dashed lines).

[Figure]

Figure R4. Comparison of the vertical profiles of velocity ($u$) predicted by the COAWST employing (a) *Rh* model using actual and modeled root projected area density profile ($a_{root}$), (b) cylinder model with sparse and (c) dense array, (d) Xie root model, (e) increased bed roughness as an approximation of vegetation drag, and (f) without imposing vegetation drag (no vegetation), and measurement by Yoshikai et al. (2022) for some selected tidal phases during the measurement period. Root mean square error (RMSE) and $R^2$ values of the modeled $u$ against the measured data are also shown, for which computation of the predicted value at the height of the measurement point was obtained by the interpolation of $u$ computed at adjacent vertical layers.

[Figure]

Figure R5. Time-series of measured and predicted (a, d) cross-sectional mean velocity ($U$), (b, e) (spatially averaged) velocity at $z = 0.05$ m, and (c, f) bed shear stress ($\tau_{bed}$) during the two-days measurement in Bakhawan Ecopark. The measured values are from Yoshikai et al. (2022) and the predicted values are obtained through the COAWST employing the Xie root model and the increased bed roughness as an approximation of drag by mangroves, respectively.

**Reviewer [1.8]:**

line 91: the reference seems to be the data of the paper. I would like to know what is different between the implemented model and the model you are referring to here

**Response [1.8]:**

The reference here (Yoshikai et al., 2021) is the paper presenting an empirical model for the *Rhizophora* root system structures (referred to as *Rh*-root model in this manuscript), which we implemented in the COAWST in this study. Therefore, the model we are referring to here is the same as the one implemented in the COAWST.

**Reviewer [1.9]:**

line 181: are you defining tree sizes as a distribution?

**Response [1.9]:**

We did not impose spatial variations in tree sizes in the model testing performed in this study, as described in L. 217 in the original manuscript. However, the model presented in this study has the capability of accounting for the spatially variable tree parameters (by defining variable stem diameter and tree density in each grid) which may be needed in a large-scale, such as a forest-scale, flow simulations in a mangrove forest as discussed in L. 463–474 in the original manuscript.

**Reviewer [1.10]:**

table 2: please make more clear that these are the measurements by linking them to the map

**Response [1.10]:**

We added a sentence in the caption of Table 2 (now Table 1 in the revised manuscript) referring to Fig. R1c (Fig. S2c in the revised manuscript) for where the hydrodynamic and vegetation parameters came from; we also added a sentence that the values for the flume experiments in Table 2 were converted to the real-scale:

> L. 257–259 in the marked-up version: "Figure S2 shows the location where the values of vegetation and hydrodynamic variables in the table were derived in Yoshikai et al. (2022a). Note that the values of vegetation and hydrodynamic variables in the flume in Maza et al. (2017) were converted to the real-scale."

Please note that the row of "Bottom roughness ($z_0$, m)" was moved to a new table (Table 2 in the revised manuscript) because we added a new test case that varied $z_0$ value (increased bed roughness case).

**Reviewer [1.11]:**

line 259: please present the sensitivity runs in the supplementary

**Response [1.11]:**

We provided the results of sensitivity runs of varying $\gamma$ in the prediction of the vertical profile of turbulent kinetic energy in Fig. S6. We included Fig. S6 in the Supporting Information as Fig. S3.

[Figure]

Figure R6. Root mean square error (RMSE) of modeled turbulent kinetic energy ($k$) against the measured data in (a) Exp 1, (b) Exp2, and (c) both Exp 1 and 2 of the flume experiment, by varying the value of scale coefficient ($\gamma$), for which the computation of the predicted value at the height of the measurement point was obtained by the interpolation of $k$ computed at adjacent vertical layers.

**Reviewer [1.12]:**
fig 4: maybe remind the reader in the caption what is HRmax

**Response [1.12]:**
We added a sentence in the caption of Fig. 4 explaining that $HR_{max}$ stands for the maximum root height (L. 392 in the marked-up version).

**Reviewer [1.13]:**
Fig 5: what absolute water levels and time-steps throughout the tidal cycle are presented here? I am not sure what the difference is between actual and modeled aroot. Is one the data and one the predictions of the implemented model? Add R^2 values to quantify the error

**Response [1.13]:**
We have provided the information on water levels in Table R2 (included in the revised manuscript as Table S5) in the analysis shown in Fig. 5. The model was run with a time-step of 2 seconds. However, please note that although the model was compared with the time-series data of the tidal cycle, we have created a steady state of flow in the model for the comparison with each measured data as described in L. 222–223 in the original manuscript and in Response [1.2].

It is correct that the $Rh$ model with actual $a_{root}$ used the measured data of root projected area density ($a_{root}$) measured in Yoshikai et al. (2022) while the $Rh$ model with modeled $a_{root}$ used the predicted $a_{root}$ by the implemented $Rh$-root model, as described in L. 283–284 in the

original manuscript. We have made this point clearer in Table R5 which is included in the revised manuscript as Table 2; please see Response [2.5] below.

Finally, we have added $R^2$ and RMSE values in the figure. Please see Fig. R4 shown above or Fig. 5 in the revised manuscript.

**Reviewer [1.14]:**

I hope that the authors can extent their analyses and revise the manuscript as I believe this will be a very useful contribution.

**Response [1.14]:**

We thank the reviewer again for the constructive comments. We believe that the revisions made have improved the quality of the manuscript substantially.

**References**

King, A. T., Tinoco, R. O., & Cowen, E. A. (2012). A $k$–ε turbulence model based on the scales of vertical shear and stem wakes valid for emergent and submerged vegetated flows. Journal of Fluid Mechanics, 701, 1–39. https://doi.org/10.1017/jfm.2012.113.

Maza, M., Adler, K., Ramos, D., Garcia, A. M., & Nepf, H. (2017). Velocity and drag evolution from the leading edge of a model mangrove forest. Journal of Geophysical Research: Oceans, 122(11), 9144–9159. https://doi.org/10.1002/2017JC012945.

Nepf, H. M. (2012). Flow and transport in regions with aquatic vegetation. Annual review of fluid mechanics, 44, 123–142. https://doi.org/10.1146/annurev-fluid-120710-101048.

Xie, D., Schwarz, C., Brückner, M. Z., Kleinhans, M. G., Urrego, D. H., Zhou, Z., & Van Maanen, B. (2020). Mangrove diversity loss under sea-level rise triggered by bio-morphodynamic feedbacks and anthropogenic pressures. Environmental Research Letters, 15(11), 114033. https://doi.org/10.1088/1748-9326/abc122.

Yoshikai, M., Nakamura, T., Bautista, D. M., Herrera, E. C., Baloloy, A., Suwa, R., Basina, R., Primavera-Tirol, Y. H., Blanco, A.C., & Nadaoka, K. (2022) Field measurement and prediction of drag in a planted *Rhizophora* mangrove forest. Journal of Geophysical Research: Oceans, 127, e2021JC018320. https://doi.org/10.1029/2021JC018320.

Yoshikai, M., Nakamura, T., Suwa, R., Argamosa, R., Okamoto, T., Rollon, R., ... & Nadaoka, K. (2021). Scaling relations and substrate conditions controlling the complexity of *Rhizophora* prop root system. Estuarine, Coastal and Shelf Science, 248, 107014. https://doi.org/10.1016/j.ecss.2020.107014.

**Response to Reviewer #2**

**Reviewer [2.1]:**
**General Comments**

This manuscript presents a new approach to modeling the flow of water within Rhizophora mangroves. The key improvements to the COAWST vegetation package are: (1) allowing the vertical varying projected area density (frontal area per unit plan area), (2) using the root and stem length-scales in the turbulence dissipation terms, (3) implementing the Rhizophora module which can calculate projected area density from easily obtainable field measurements. These improvements allow the field to move beyond the conventional cylinder assumption, and are generally applicable to all hydrodynamically rough environments which aren't well described by cylinders.

I liked the approach and theme of this paper, and think with some revisions it would be a nice contribution. I also really appreciated the detail of the supplemental information.

**Response [2.1]:**
We thank the reviewer for the thorough assessment and constructive comments on our manuscript. Please see our responses to the comments below.

**Reviewer [2.2]:**
**Specific Comments**

1. The no vegetation case shown in Fig. 5 and Fig S1.

   1. While including this test case in the manuscript is interesting to see how the hydrodynamics are changed if the mangroves were removed from the ecosystem, it seems to change direction from the rest of the paper. I believe the message of this paper is comparing how this new approach of accounting for the roughness of mangroves is different from past approaches (cylinder arrays or enhanced z0 values). I think that if the z0 value for the no-vegetation case is increased, maybe similarly to the Zhang 2012 mentioned on line 65, this would fit with the theme of figures 5 and 6 which contrast the new approach to past approaches.

**Response [2.2]:**
We thank the reviewer for the suggestion. Because we think that the results of the no vegetation case shown in Fig. 5 and Fig. S1 (now Fig. S4 in the revised manuscript) are important for demonstrating how much the drag by mangrove forests could be significant in affecting the flows in mangrove forests, hence strengthening the importance of proper parameterization of the impacts of mangroves, we would like to keep them in the manuscript.

We agree with the reviewer that a case with increased bed roughness ($z_0$ value) would fit the theme of the paper and add some insights into the effects of different drag parameterization.

Below, we describe how the bed shear stress is computed in the COAWST, and how the model is tested using the increased $z_0$ value as an approximation of mangrove drag.

In the COAWST, bed shear stress is computed based on quadratic law using the velocities at the bottom computational cell as (Warner et al., 2008):

$$\tau_{bed} = \rho_w C_{bed} u^2 \tag{R3}$$

where $\tau_{bed}$ is the bed shear stress (N m$^{-2}$), $\rho_w$ is the water density (kg m$^{-3}$), $C_{bed}$ is the bed drag coefficient, and $u$ is the flow velocity (m s$^{-1}$) computed at the bottom cell. It assumes that the flow in the bottom boundary layer has the classic vertical logarithmic profile as:

$$|u| = \frac{u_*}{\kappa} \ln\left(\frac{z_{bottom}}{z_0}\right) \tag{R4}$$

where $u_*$ is the friction velocity, $\sqrt{\tau_{bed}}$, $\kappa = 0.41$ is the von Kármán constant, $z_{bottom}$ is the mid-elevation point of the bottom computational cell above the bed (m), and $z_0$ is the bed roughness length (m). From Eqs. (R3)–(R4), the $C_{bed}$ is calculated using $z_0$ as:

$$C_{bed} = \kappa^2 \left[\ln\left(\frac{z_{bottom}}{z_0}\right)\right]^{-2} \tag{R5}$$

The value of $z_0$ or $C_{bed}$ can be related to the Manning's coefficient ($n_{manning}$) as follows considering turbulent open channel flow. In an open channel flow with depth-averaged velocity $U_{mean}$, water depth $h$, and bed slope $S_0$, the $U_{mean}$ can be described using the Manning's coefficient as:

$$U_{mean} = \frac{1}{n_{manning}} h^{2/3} S_0^{1/2} \tag{R6}$$

Assuming a steady flow where the momentum balance can be reduced to an equilibrium between the bed shear stress $\tau_{bed}$ and the gravitational (or pressure) forces driving the flow, the bed shear stress can be expressed as (Crompton et al., 2020):

$$\tau_{bed} = \rho_w g h S_0 \tag{R7}$$

where $g$ is the gravitational acceleration (m s$^{-2}$). From Eq. (R6)–(R7) and assuming that the depth-averaged form of Eq. (R3), $\tau_{bed} = \rho_w C_{bed,mean} U_{mean}^2$, is valid, the Manning's coefficient can be expressed as:

$$n_{manning} = h^{1/6} \sqrt{\frac{C_{bed,mean}}{g}} \tag{R8}$$

where $C_{bed,mean}$ is the bed drag coefficient which is used for computing $\tau_{bed}$ using the $U_{mean}$. Also, by relating the depth-averaged form of Eq. (R5), $C_{bed,mean}$ can be expressed using $z_0$ as (Lenz et al., 2017):

$$C_{bed,mean} = \kappa^2 \left[\ln\left(\frac{h}{z_0}\right)\right]^{-2} \tag{R9}$$

Considering the Manning's coefficient of 0.14, which is a value typically used for approximating the drag by mangroves (e.g., Zhang et al., 2012), and a water depth of 0.5 m, based on Eqs. (R8)–(R9), the equivalent bed roughness $z_0$ is 0.22 m.

As suggested by the reviewer, we performed an additional model analysis using an increased $z_0$ value as an approximation of mangrove drag. However, the application of Eqs. (R3)–(R5) needs the condition $z_0 < z_{bottom}$, which limits the applicable $z_0$ value depending on the water depth or thickness of the bottom cell. In order to increase the applicable $z_0$ value in our analysis, where the lowest water depth examined was around 0.15 m (Fig. 6; Table R3), we reduced the number of vertical layers from 5 to 3, which increased the minimum $z_{bottom}$ up to 0.025 m. We then conducted the analysis using $z_0 = 0.02$ m as a case of increased $z_0$; however, this value is considered lower compared to the typical Manning's coefficient value of 0.14 (of which the equivalent value is $z_0 = 0.22$ m when the water depth is 0.5 m)

Results of the model analysis with the increased $z_0$ are provided in Figs. R4e and R5 (Figs. 5e and 8 in the revised manuscript), which can be seen below. The model predicted the significant attenuation of flow velocity from the surface to the bottom due to the large bottom friction produced by the increased $z_0$, which did not well represent the actual conditions of the velocity profile in the *Rhizophora* mangrove forest (Fig. R4e).

Comparison with the time-series data showed contrasting results with the one using the Xie root model (please see Response [1.7] for the descriptions of the case using Xie root model). Although both cases showed a large overestimation of flow velocity when the water depth is relatively high (e.g., $h > 0.3$ m), the predicted $U$ in the case using the increased $z_0$ approached the measured data with decreasing water depth while the case using the Xie root model consistently overestimated the $U$ throughout the measurement period (Fig. R5a–e). This different trend is due to the different drag parameterization with the bed roughness or objects within the water column. Specifically, the decrease in the total projected area of submerged part of the objects exerting drag with a decrease in water depth cannot be accounted for by the mangrove drag approximation with the bed drag. As a consequence, the bed drag became significant in decelerating the flow velocity when the water depth is low ($h < 0.3$ m) compared to the case using the Xie root model, as seen in the equivalent $U$ to the measured values (Fig. R5a, d). Because bed drag is the main force to counteract the imposed pressure gradient in the increased $z_0$ case, the bed drag showed the overestimation as expected (Fig. R5c, f). This will lead to large overestimation of sediment erosion, suggesting that increased bed roughness does not work well for simulating the sediment transport in mangrove forests.

In the revised manuscript, we have included the condensed version of the above descriptions as followings:

- Text S7 in the Supporting Information: The description of the bed shear stress calculation in the COAWST and how the value of $z_0$ was chosen for a case of increased bed roughness.

- L. 332–333, 343–345 in the marked-up version (Section 2.2.2): The description on the increased bed roughness case for model testing
- L. 408–410, 439–444 in the marked-up version (Section 3.2): The description on the results of the increase bed roughness
- L. 506–513 in the marked-up version (Section 4.1): The discussions on the results of the increased bed roughness
- Figure 5: Replaced with Fig. R4.
- Figure 8: Replaced with Fig. R5

Additionally, given the increase in the discussions on the model performance, Section 4.1 "Model performance" in the original manuscript was split into two sections Section 4.1 "Performance of the previously proposed drag parameterization" and Section 4.2 "Performance of the new model".

Also, as described in Response [1.7], we have updated the model code and created new files for running the new model test cases, thus we have issued a new DOI for the updated model code and data archived in Zenodo. Accordingly, the "Code and data availability" section (L. 672 in the marked-up version) was revised.

[Figure]

Copy of Figure R4 shown above in Response [1.7]: Comparison of the vertical profiles of velocity ($u$) predicted by the COAWST employing (a) $Rh$ model using actual and modeled root projected area density profile ($a_{root}$), (b) cylinder model with sparse and (c) dense array, (d) Xie root model, (e) increased bed roughness as an approximation of vegetation drag, and (f) without imposing vegetation drag (no vegetation), and measurement by Yoshikai et al. (2022) for some selected tidal phases during the measurement period. Root mean square error

(RMSE) and R² values of the modeled *u* against the measured data are also shown, for which computation of the predicted value at the height of the measurement point was obtained by the interpolation of *u* computed at adjacent vertical layers.

[Figure]

Copy of Figure R5 shown above in Response [1.7]: Time-series of measured and predicted (a, d) cross-sectional mean velocity (*U*), (b, e) (spatially averaged) velocity at *z* = 0.05 m, and (c, f) bed shear stress ($\tau_{bed}$) during the two-days measurement in Bakhawan Ecopark. The measured values are from Yoshikai et al. (2022) and the predicted values are obtained through the COAWST employing the Xie root model and the increased bed roughness as an approximation of drag by mangroves, respectively.

**Reviewer [2.3]:**

2. Does the sparse cylinder model (line 289) have an equivalent frontal area to the field data? Similarly to how the cylinder models used in EXP1 and EXP2 have equivalent frontal area to the Rh model. If so, then this section would nicely flow from the lab section where frontal area was conserved. If not, then there is a jump in the methods being used to create the cylinder arrays in the lab vs in the field section. For best transition between the sections I think the method for generating the cylinder arrays should be consistent between the lab and field parts of the paper.

**Response [2.3]:**

The cylinder model applied for the field-based study does not have the equivalent frontal area to the field data in contrast to the cylinder models applied for Exp1 and Exp2 of the laboratory-based study.

We would like to note that the objectives of the model applications to the laboratory- and field-based studies are different. The main objective of the application to the laboratory-based study is to explore the effectiveness of the formulations for the drag and turbulence terms (Eqs. (1)–(6)), which were newly implemented in the COAWST in this study, compared to the cylinder drag model provided the vegetation frontal area density ($a$) as a known parameter. Alternatively, in the case of field studies, the parameter $a$ is usually unknown and needs to be predicted for the model application. Thus, one of the main objectives of the application to the field-based study is to explore the effectiveness of the implemented *Rhizophora* root model – the predictor of $a$ – in combination with the new formulations in the COAWST, compared to the drag parameterizations proposed in previous studies.

The vegetation frontal area density ($a$), which is a labor-intensive parameter to obtain in the field, has remained as a factor making the model application to the field mangrove forests challenging. This is why several modeling studies have parameterized the drag by mangroves in different ways such as cylinder array approximation with arbitrary cylinder density as in Xie et al. (2020) (described in L. 64–66 in the original manuscript), cylinder array approximation based on the vegetation geometry measured at a height of around 0.25 m in Horstman et al. (2013) (described in L. 288–289 in the original manuscript), and increased bed roughness in Zhang et al. (2012) (described in L64–66 in the original manuscript). In this study, we have examined these drag parameterizations for the application to the field mangrove forest by comparing them with the new model presented in this study, rather than defining the cylinder array having an equivalent frontal area to the field data, assuming that the parameter $a$ is unknown (please note that we have examined the parameterization of Zhang et al. (2012) in the form of dense cylinder arrays – please see L. 289–291 in the original manuscript; we have also added new simulation cases using the root model of Xie et al. (2020) and the increased bed roughness – please see Response [1.7] and Response [2.2], respectively).

In the revised manuscript, we included a paragraph describing this point, specifically the difference in the objectives of model applications to the laboratory- and field-based studies (L. 246 – 254 in the marked-up version).

**Reviewer [2.4]:**

3. I would love to see a figure that shows the difference accounting for 2 length-scales in the turbulence routines makes. This is mentioned on lines 444-445 and lines 424-426. I haven't seen a figure that does this yet, and the changes to the code are already made so I think this could be a nice addition to the paper.

**Response [2.4]:**

As suggested by the reviewer, we have examined the effects of accounting for the different length-scales of stem- and root-generated wakes in the presented model (*Rh* model). We have done it by performing additional model analyses for the flume experiments that set the two length-scales to either the root diameter ($D_{root,ave}$) or the stem diameter ($D_{stem,ave}$).

The results clearly showed that without accounting for the two different length-scales, the model fails to predict the vertical profile of turbulent kinetic energy ($k$) (Fig. R7). If the two length-scales are set to the stem diameter, the model largely overestimates $k$ specifically in the root zone ($z/HR_{max} < 1$), whereas the model with the length-scales set to the root diameter largely underestimated $k$ in the upper and above the root zone ($z/HR_{max} > 0.5$). The model that has the length-scales set to the root diameter showed a slight increase in $k$ at the height around $z/HR_{max} = 0.9$ (Fig. R7b), possibly due to the velocity shear generated by the sharp decrease in frontal area at the top of the root zone (Maza et al., 2017; Fig. 3a). However, this increase in $k$ alone is not enough to explain the significantly higher $k$ at upper and above the root zone compared to the lower root zone, suggesting the significant roles of the different length-scales of stem- and root-generated wakes in shaping the overall structure of $k$.

In the revised manuscript, we have included Fig. R7 in the Supporting Information as Fig. S5, and referred to it in L. 543–545 in the Section 4.2 (marked-up version) as:

> "Without accounting for the two different length-scales, the model failed to reproduce the TKE profile while the velocity profile remained similar, suggesting the minor importance of shear production for reproducing the TKE (Fig. S5)."

[Figure]

Figure R7. Comparison of the vertical profiles of (temporally and spatially averaged) velocity ($u$) and turbulent kinetic energy ($k$) normalized by the cross-sectional mean velocity ($U$) measured by Maza et al. (2017) and predicted by the COAWST using the $Rh$ model with different length-scales of stem- and root-generated wakes ($L_{stem}$ and $L_{root}$, respectively) defined – blue markers: $L_{stem}$ and $L_{root}$ set to the stem diameter ($D_{stem,ave}$) and root diameter ($D_{root,ave}$), respectively; dark-gray markers: $L_{stem}$ and $L_{root}$ both set to $D_{root,ave}$; light-gray markers: $L_{stem}$ and $L_{root}$ both set to $D_{stem,ave}$. The scale coefficient ($\gamma$) was set to 1.2 for all the cases.

**Reviewer [2.5]:**

4. I would appreciate if the cylinder metrics (diameter, density, height) mentioned in EXP1,EXP2, the sparse cylinder array and the dense cylinder array could be compiled into a table and attached as supplemental information.

    1. I believe it is important to state the height of these cylinder arrays. Based on the velocity profiles shown in Figures 4 and 5, I believe all the cylinder arrays span the entire water column. However there are other papers which use cylinder arrays which span a fraction of the water column, so I think it is important to state.

    2. The widths are also important to state because of they are used in the turbulence dissipation term. The diameters for EXP1 and EXP2 are already stated, but I couldn't find diameters for the sparce and dense cylinder arrays.

**Response [2.5]:**

We have compiled the parameter settings of all the simulation cases in Table R5. Because we consider that Table R5 is convenient for the readers to grasp the different model configurations used in this study, we have included it in the main text of the revised manuscript as Table 2.

Regarding comment 1, the height of the cylinder arrays was defined well higher than the water level in the model, thus they span the entire water column as pointed out by the reviewer. In the revised manuscript, we have added an explanation of this point in Section 2.2.1 as

> L. 307–309: "Cylinder height was set well higher than the water level to create the condition that cylinders span the entire water column – this also applies to the cylinder drag model examined in the next section."

Regarding comment 2, we noticed that we missed adding the information on the cylinder diameter defined for the field-based study in the original manuscript. We have provided the information on the cylinder diameter in Table R5, which have been included in the revised manuscript as Table 2.

Table R5. Tested model configurations to represent the impact of *Rhizophora* mangroves against flume experiments (Exp1 and 2) in Maza et al. (2017) and field measurement in Yoshikai et al. (2022a). $n_{tree}$: tree density; $n_v$: cylinder density; $D_{stem,ave}$: mean stem diameter; $b_v$: cylinder density; $a_{root}$: root projected area density; $D_{root,ave}$: mean root diameter; $z_0$: bed roughness length; $N_{layer}$: number of vertical layers of model grid.

| Test case | Model configuration | Parameter settings | | | | | |
|---|---|---|---|---|---|---|---|
| | | $n_{tree}$ or $n_v$ (m$^{-2}$) | $D_{stem,ave}$ or $b_v$ (m) | $a_{root}$ (m$^{-1}$) | $D_{root,ave}$ (m) | $z_0$ (m) | $N_{layer}$ |
| Flume experiment | *Rh* model | 0.072 ($n_{tree}$) | 0.2 ($D_{stem,ave}$) | Measured value [a] | 0.038 | 0.5 × 10$^{-3}$ [e] | 15 |
| | Cylinder model for Exp1 | 1.22 ($n_v$) | 0.038 ($b_v$) | - | - | 0.5 × 10$^{-3}$ [e] | 15 |
| | Cylinder model for Exp2 | 1.76 ($n_v$) | 0.038 ($b_v$) | - | - | 0.5 × 10$^{-3}$ [e] | 15 |
| Field measurement | *Rh* model with actual $a_{root}$ | 0.36 ($n_{tree}$) | 0.066 ($D_{stem,ave}$) | Measured value [b] | 0.030 | 0.5 × 10$^{-3}$ | 5 |

| | | | | | | |
|---|---|---|---|---|---|---|
| *Rh* model with modeled $a_{root}$ | 0.36 ($n_{tree}$) | 0.066 ($D_{stem,ave}$) | Modeled value [c] | 0.030 | $0.5 \times 10^{-3}$ | 5 |
| Cylinder model (sparse) | 13.5 ($n_v$) | 0.030 ($b_v$) | - | - | $0.5 \times 10^{-3}$ | 5 |
| Cylinder model (dense) | 32.3 ($n_v$) | 0.030 ($b_v$) | - | - | $0.5 \times 10^{-3}$ | 5 |
| Xie root model | 0.36 ($n_{tree}$) | 0.066 ($D_{stem,ave}$) | Eq. (R2) [d] | 0.010 | $0.5 \times 10^{-3}$ | 5 |
| Increased $z_0$ | - | - | - | - | 0.02 | 3 |
| No vegetation | - | - | - | - | $0.5 \times 10^{-3}$ | 5 |

[a] Corresponds to the value of black markers minus $n_{tree}D_{stem,ave}$ in Fig. 3a.
[b] Corresponds to the value of black markers minus $n_{tree}D_{stem,ave}$ in Fig. 3b.
[c] Corresponds to the value of blue markers minus $n_{tree}D_{stem,ave}$ in Fig. 3b.
[d] Corresponds to the value of light green markers minus $n_{tree}D_{stem,ave}$ in Fig. 3b.
[e] Assumed value.

**Reviewer [2.6]:**

5. Lines 200-211, I think that this section and table 1 can be removed from the paper or moved to supplemental information. This section might be useful as a guide for someone using your code, but I don't think it adds much value towards understanding the content of the manuscript.

**Response [2.6]:**

We agree with the reviewer. We have moved the said section and Table 1 to the Supporting Information as Text S5 "Implementation of the new model to the COAWST" and Table S2 in the revised manuscript.

**Reviewer [2.7]:**

6. Generally it seems the core modifications to the code are in the drag term and the turbulence routines, and these change the flow field which then might change the sedimentation rates. I think the sediment parts of this paper are a case study of how these model changes can affect a variable of interest (like sedimentation rates), but I don't think the focus of the paper should be on the changes in sedimentation rates.

**Response [2.7]:**

We agree with the reviewer that the simulations of the sediment transport should not be the focus of the paper. To underscore the contribution of this manuscript, we have removed the section in the materials & methods, results, and discussion on the sediment transport simulation in the revised manuscript. Please also see Response [1.1], [1.3], and [1.5] in relation to this point. Changes made to the manuscript related to this point are listed in Response [1.1]. We believe that this revision made the manuscript more focused on the novelty and the contribution of this study.

**Reviewer [2.8]:**

**Technical corrections/ Typing errors**

1. Line 308 "run" should be "ran"

2. Line 214, could you please consider stating the vertical resolution and/or the number of sigma levels used? This is mentioned in line 241, but I think it would be nice to have all of the domain characteristics mentioned in the same place.

**Response [2.8]:**

The sentence of L. 308 in the original manuscript was removed in the revised when removing the Section 2.3 "Sediment transport simulation".

We included the information on the vertical layering in L. 236–238 (marked-up version) as

"We set 15 vertical layers with approximately uniform layer thickness to be applied to the laboratory-based study. For the field-based study, the number of vertical layers was reduced to 5 because of the shallow water depths."

Accordingly, the same information which had been written in the Sections 2.2.1 and 2.2.2 in the original manuscript was removed in the revised manuscript (L. 284 and 324–325 in the marked-up version). In addition, we have also included the information on the number of vertical layers in Table 2 in the revised manuscript.

**References**

Crompton, O., Katul, G. G., & Thompson, S. (2020). Resistance formulations in shallow overland flow along a hillslope covered with patchy vegetation. Water Resources Research, 56(5), e2020WR027194. https://doi.org/10.1029/2020WR027194.

Horstman, E., Dohmen-Janssen, M., & Hulscher, S. J. M. H. (2013, June). Modeling tidal dynamics in a mangrove creek catchment in Delft3D. In Coastal dynamics (Vol. 2013, pp. 833–844).

Lentz, S. J., Davis, K. A., Churchill, J. H., & DeCarlo, T. M. (2017). Coral reef drag coefficients–water depth dependence. Journal of Physical Oceanography, 47(5), 1061-1075.

Maza, M., Adler, K., Ramos, D., Garcia, A. M., & Nepf, H. (2017). Velocity and drag evolution from the leading edge of a model mangrove forest. Journal of Geophysical Research: Oceans, 122(11), 9144–9159. https://doi.org/10.1002/2017JC012945.

Warner, J. C., Sherwood, C. R., Signell, R. P., Harris, C. K., & Arango, H. G. (2008). Development of a three-dimensional, regional, coupled wave, current, and sediment-transport model. Computers & geosciences, 34(10), 1284–1306. https://doi.org/10.1016/j.cageo.2008.02.012.

Xie, D., Schwarz, C., Brückner, M. Z., Kleinhans, M. G., Urrego, D. H., Zhou, Z., & Van Maanen, B. (2020). Mangrove diversity loss under sea-level rise triggered by bio-morphodynamic feedbacks and anthropogenic pressures. Environmental Research Letters, 15(11), 114033. https://doi.org/10.1088/1748-9326/abc122.

Yoshikai, M., Nakamura, T., Bautista, D. M., Herrera, E. C., Baloloy, A., Suwa, R., Basina, R., Primavera-Tirol, Y. H., Blanco, A.C., & Nadaoka, K. (2022) Field measurement and prediction of drag in a planted Rhizophora mangrove forest. Journal of Geophysical Research: Oceans, 127, e2021JC018320. https://doi.org/10.1029/2021JC018320.

Zhang, K., Liu, H., Li, Y., Xu, H., Shen, J., Rhome, J., & Smith III, T. J. (2012). The role of mangroves in attenuating storm surges. Estuarine, Coastal and Shelf Science, 102, 11–23. https://doi.org/10.1016/j.ecss.2012.02.021.

---

## Referee Report (RR1)

I read the manuscript by Yoshikai et al. with great pleasure and believe that it will be suitable for publication very soon. I thank the authors for addressing my comments and the additional work that they carried out to improve the manuscript. I think the manuscript reads much clearer now and the methods are very clear! However, I have still some suggestions to improve the manuscript and get it ready for publication. My main concern is that the abstract and introduction require framing of the study and presentation of the questions that the manuscript addresses. For example, I think the attention is still focused too much on the role of sedimentation in wetlands. This distracts from the key messages and the novel work that the authors do, so I suggest to streamline the first third of the paper more towards the hydrodynamics (whose representation is by itself very important and therefore represents a significant contribution). In addition, I suggest to have another read of the text as there are still some ambiguities and repetitions that could be removed to make the manuscript even clearer. For example, sentences are generally very long and convoluted which makes the text sometimes hard to follow (especially in the conclusions). I included many textual suggestions in the pdf that hopefully are of some use to the authors.

Another point is the use of the Xie-model. Although I think it is very interesting to compare the presented model to another model, I would be careful with the interpretation of the scenario of the Xie-model, as Xie et al have a spatially and temporally varying stem diameter and density. So, the combined spatial and temporal evolution of the mangrove forest determines the mean hydrodynamics and sediment transport and deposition processes over long time-scales. Here only one scenario with constant diameter and density is tested for two points in time. I think the results can still be presented as is, but I suggest to rename the scenario with a more general term and bring up Xie et al as an example study that uses this type of drag representation.

**General comments:**

Abstract:

I think the abstract is still very much focussed on the sedimentary processes, which of course are important but here the hydrodynamics are the main focus. I would possibly adjust the text to focus on the fact that we need to better represent the hydrodynamics to in turn better describe the sedimentary processes. For example, the first sentence as it is very general now and could be more streamlined towards mangroves and hydrodynamics instead of sedimentation and transport and sea level rise. I also have several suggestions to make the abstract more concise and less vague in the pdf.

Introduction:

Again, I suggest to focus more on the hydrodynamics and less on the link with sediment and geomorphology (e.g., in lines 71, 79)

line 90: At this point I still was not sure what the differences is between the presented model and the empirical model by Yoshikai et al 2021. The latter is mentioned late in the introduction and it is not entirely clear why both are used. Maybe explain the model upfront and what both scenarios are trying to answer?

The paragraph starting in line 72 seems like it could be removed or maybe you could explain here the empirical model?

Methods:

I believe that a map of the sites would be useful here in the main manuscript. In Figure 2 the names (Bak 1,2, and Fuk) are now not clear and I was not sure where to find the site description. Figure S2 also does not have labels with those names.

Results:

In the introduction and methods first velocity is presented and then TKE. In the results (paragraphs in lines 317 and 329) it is the other way around. I suggest to swap the two paragraphs to be consistent with the structure.

Conclusions:

Sentences are very long in the conclusions. I also suggest to make the aim of the study clearer in the first sentence.

**Minor comments:**

I suggest to revisit the text and make sure it is correct and clear. Below some suggestions of what I think could be rephrased but there are other instances, so please have a thorough look.

line 49: the part with the reference of Nepf et al seems a bit odd.

line 95: a very long convoluted sentence that could be rephrased.

line 97: "We modified the equations introduced by Beudin et al. (2017) to make them suitable for representing the impact of Rhizophora mangroves on flow; these equations are described below. We added a new module in COAWST–Rhizophora root module–that provides the vertical profile of the projected area density of root systems from stem diameter and tree density in each model grid (Fig. 1)." Is the second sentence what you did in the first sentence or did you do two steps here?

line 110: "The Reynolds number …" – I am not sure if this was found in the publication (Shan et al) or in your work. Please clarify.

line 301: " We inputted …" Could this sentence be combined with the previous one? Now it seems like the same thing is said twice.

line 450: very long sentence. Please separate into two or three sentences.

line 463: again very long, please separate.

line 485: I think you can leave out the first sentence. As a reader I would not expect you to do more work so discussing what can be done in the future suffices here.

line 504: I would acknowledge here studies that use dynamic vegetation models (which you cited in the foregoing line)

[revised manuscript text omitted]

---

## Referee Report (RR2)

**General Comments**

This manuscript presents a new approach to modeling the flow of water within Rhizophora mangroves. The key improvements to the COAWST vegetation package are: (1) allowing the vertical varying projected area density (frontal area per unit plan area), (2) using the root and stem length-scales in the turbulence dissipation terms, (3) implementing the Rhizophora module which can calculate projected area density from easily obtainable field measurements. These improvements allow the field to move beyond the conventional cylinder assumption, and are generally applicable to all hydrodynamically rough environments which aren't well described by cylinders.

I like the approach of this paper. The changes the authors have made have increased the clarity, and strength of this paper.

**Comment on Response 2.2**

The runs that are labeled as increased bed roughness (z0=0.02) should be considered with care because the z0 value used in those runs are an order of magnitude less than the authors estimate of the actual (without numerical limitations) increased bed roughness (z0=0.22). I think some text in the manuscript describing z0=0.2 as the maximum amount of bed roughness that logarithmic drag can represent in the model due to numeric limitation would be good. The authors thoroughly explain the numerical limitation in the supplemental information, I believe a few words in the manuscript would make it very clear to readers that z0=0.02 isn't the authors estimate of the enhanced z0 value. Alternatively, I believe that the requirement in COAWST that z0 < zbottom, is only true when using a logarithmic drag law. I have never attempted this, but I think it would be possible to use the equation (R8) relating the manning coefficient and $C_{\{bed,mean\}}$ to arrive at a drag coefficient that can be input to COAWST using a quadratic drag law, getting around the z0 < zbottom limitation. Either additional text detailing the numerical limitations of and enhanced z0 value or a quadratic drag law approach would be sufficient in addressing this.

**Comments on Responses 2.3-2.8**

I agree with the author's responses to comments 2.3-2.8.

---

## Author Response (AR2)

Dear editor and reviewers,

We thank the reviewers for taking the time to review our manuscript and providing constructive comments.

Below, we address the reviewers' comments point by point. In the response, black- and blue-colored characters denote reviewers' comments and our responses, respectively.

**Response to Reviewer #1**

**Reviewer [1.1]:**
I read the manuscript by Yoshikai et al. with great pleasure and believe that it will be suitable for publication very soon. I thank the authors for addressing my comments and the additional work that they carried out to improve the manuscript. I think the manuscript reads much clearer now and the methods are very clear! However, I have still some suggestions to improve the manuscript and get it ready for publication. My main concern is that the abstract and introduction require framing of the study and presentation of the questions that the manuscript addresses. For example, I think the attention is still focused too much on the role of sedimentation in wetlands. This distracts from the key messages and the novel work that the authors do, so I suggest to streamline the first third of the paper more towards the hydrodynamics (whose representation is by itself very important and therefore represents a significant contribution).

**Response [1.1]:**
We are grateful to the reviewers for taking the time to review our manuscript again and providing valuable comments.

We agree with the reviewer's suggestion that the manuscript needs framing of the study in the abstract and introduction focusing on the importance of hydrodynamics.

In the revised manuscript, we have reduced the descriptions related to sedimentation in wetlands in the abstract, introduction, and conclusions. Instead, we made it clearer that the focus and contribution of the study is improved modeling of hydrodynamics in *Rhizophora* mangrove forests.

The revision related to this point can be found in the following lines in the marked-up version:

- Abstract: L. 15–20; L21–22; L. 31–33
- Introduction: L. 42–56; L. 80–82; L. 95–96
- Conclusions: L. 550–552

**Reviewer [1.2]:**
In addition, I suggest to have another read of the text as there are still some ambiguities and repetitions that could be removed to make the manuscript even clearer. For example,

sentences are generally very long and convoluted which makes the text sometimes hard to follow (especially in the conclusions). I included many textual suggestions in the pdf that hopefully are of some use to the authors.

**Response [1.2]:**

We thank the reviewer for the text suggestions. We have incorporated them in the revised manuscript. We have also carefully read the manuscript again and made some revisions to long or ambiguous sentences (e.g., L. 27–31; L. 236–238; L. 332–339 in the marked-up version).

**Reviewer [1.3]:**

Another point is the use of the Xie-model. Although I think it is very interesting to compare the presented model to another model, I would be careful with the interpretation of the scenario of the Xie-model, as Xie et al have a spatially and temporally varying stem diameter and density. So, the combined spatial and temporal evolution of the mangrove forest determines the mean hydrodynamics and sediment transport and deposition processes over long time-scales. Here only one scenario with constant diameter and density is tested for two points in time. I think the results can still be presented as is, but I suggest to rename the scenario with a more general term and bring up Xie et al as an example study that uses this type of drag representation.

**Response [1.3]:**

We agree with the reviewer's suggestion. We have renamed the scenario from "Xie root model" to "Generic root model" throughout the text. Figs. 3, 5, 8, and S6 have also been updated to reflect this change. We have also added the following sentence in L. 334–336 (marked-up version):

> "We use the term "generic" because Xie et al. (2020) used this model to represent root structures of several different mangrove genera including *Rhizophora*."

**Reviewer [1.4]:**
**General comments:**
Abstract:
I think the abstract is still very much focussed on the sedimentary processes, which of course are important but here the hydrodynamics are the main focus. I would possibly adjust the text to focus on the fact that we need to better represent the hydrodynamics to in turn better describe the sedimentary processes. For example, the first sentence as it is very general now and could be more streamlined towards mangroves and hydrodynamics instead of sedimentation and transport and sea level rise. I also have several suggestions to make the abstract more concise and less vague in the pdf.

**Response [1.4]:**

This was addressed in Responses [1.1] and [1.2].

**Reviewer [1.5]:**
Introduction:
Again, I suggest to focus more on the hydrodynamics and less on the link with sediment and geomorphology (e.g., in lines 71, 79)

**Response [1.5]:**
This was addressed in Responses [1.1].

**Reviewer [1.6]:**
line 90: At this point I still was not sure what the differences is between the presented model and the empirical model by Yoshikai et al 2021. The latter is mentioned late in the introduction and it is not entirely clear why both are used. Maybe explain the model upfront and what both scenarios are trying to answer?

**Response [1.6]:**
We have added an explanation of the empirical model for *Rhizophora* root structures by Yoshikai et al. (2021) in the 5th paragraph (L. 83–88).

We have revised the said sentence as follows:

Previous manuscript:

"Here, we aim to examine the following: (a) how does the new representation of *Rhizophora* mangroves in the hydrodynamic model improve the predictability of flow velocity and turbulence compared to the conventional drag approximation using cylinder arrays or increased bed roughness? (b) how can the new model be effectively applied for an accurate prediction of the flow in *Rhizophora* mangrove forests by incorporation of the *Rhizophora* root model?"

Revised manuscript (L104–109 in the marked-up version):

"Here, we aim to examine the following: (a) how does the consideration of the three-dimensional root structures of *Rhizophora* mangroves in the hydrodynamic model improve the predictability of flow velocity and turbulence compared to the conventional drag approximation using cylinder arrays or increased bed roughness? (b) how can the new model be effectively applied to *Rhizophora* mangrove forests in the field with limited known root parameters?"

**Reviewer [1.7]:**
Methods:

I believe that a map of the sites would be useful here in the main manuscript. In Figure 2 the names (Bak 1,2, and Fuk) are now not clear and I was not sure where to find the site description. Figure S2 also does not have labels with those names.

**Response [1.7]:**

We have provided in Fig. R1 of this document the map of the sites.

We have provided the description of the sites in Text S4 in the Supporting Information of the original manuscript, but it was mistakenly referred to as "Text 4" (L. 201 in the marked-up version), and we believe that this caused the confusion of the reviewer.

In the revised manuscript, we included Fig. R1 in the Supporting Information as Fig. S1 in the Supporting Information and revised L. 199–200 (marked-up version) as:

> "We investigated the above assumption using tree census data collected from three sites (Bak1, Bak2, and Fuk; see Fig. S1 and Text S4 in the Supporting Information for the map and description of the sites)."

[Figure]

Figure R1. Map of the sites (Bak1, Bak2, and Fuk) indicated in Fig. 2. The white dots in panels "d" and "e" represent the tree census stations from which data are used in Fig. 2. In panel "e", the approximate locations of the 30-year-old (Bak1) and 17-year-old (Bak2) planted stands are also indicated. See Text S4 in the Supporting Information for the description of each site. Shorelines in panel "a–c" are from the Global Self-consistent, Hierarchical, High-resolution Geography (GSHHG) database. The aerial photo in panel "d" is from Asia Air Survey Co. Ltd., Japan, and the satellite image in panel "e" is from Google Earth.

**Reviewer [1.8]:**

Results:

In the introduction and methods first velocity is presented and then TKE. In the results (paragraphs in lines 317 and 329) it is the other way around. I suggest to swap the two paragraphs to be consistent with the structure.

**Response [1.8]:**

The said paragraphs already describe the results in the suggested order (velocity and then TKE), thus we think that revision is not necessary here.

We would like to note that the first paragraph describes the results from the *Rh*-model and the second paragraph describe the results from the cylinder model, but in each paragraph, the order of the description is velocity and then TKE.

**Reviewer [1.9]:**

Conclusions:

Sentences are very long in the conclusions. I also suggest to make the aim of the study clearer in the first sentence.

**Response [1.9]:**

We have revised the first two sentences in the Conclusions as follows (L. 550–552 in the marked-up version):

> "Modeling flow in *Rhizophora* mangroves has been challenging due to their complex root structures. This manuscript presents a new model to represent the impacts of *Rhizophora* mangroves on flow implemented in the COAWST towards a better understanding of hydrodynamics in mangrove forests."

We have also improved the texts in the Conclusions as mentioned in Response [1.2]. Please see the marked-up revised manuscript for the details of the revision.

**Reviewer [1.10]:**

**Minor comments:**

I suggest to revisit the text and make sure it is correct and clear. Below some suggestions of what I think could be rephrased but there are other instances, so please have a thorough look.

**Response [1.10]:**

This was addressed in Responses [1.1] and [1.2].

**Reviewer [1.11]:**

line 49: the part with the reference of Nepf et al seems a bit odd.

**Response [1.11]:**

Please see L.58 –61 in the marked-up version for the revision made to this sentence.

**Reviewer [1.12]:**

line 95: a very long convoluted sentence that could be rephrased.

**Response [1.12]:**

The sentence was separated into two as (L. 113–116 in the marked-up version):

> "The vegetation module has been added by Beudin et al. (2017) to account for the drag by vegetation (such as seagrasses and salt marshes) in the momentum equations in ROMS. The equations added by Beudin et al. (2017) are basically in the same form as the cylinder drag model (see Text S1 in the Supporting Information)."

**Reviewer [1.13]:**

line 97: "We modified the equations introduced by Beudin et al. (2017) to make them suitable for representing the impact of Rhizophora mangroves on flow; these equations are described below. We added a new module in COAWST–Rhizophora root module–that provides the vertical profile of the projected area density of root systems from stem diameter and tree density in each model grid (Fig. 1)." Is the second sentence what you did in the first sentence or did you do two steps here?

**Response [1.13]:**

These are two steps – the modification of the drag and turbulence model (described in Sections 2.1.1 and 2.1.2), and the incorporation of the empirical *Rhizophora* root model to the modified drag and turbulence model (described in Section 2.1.3).

In the revised manuscript, we have revised the said sentence as follows (L. 116–120 in the marked-up version):

> "We modified these equations to make them suitable for representing the impact of *Rhizophora* mangroves on flow; these equations are described below (Sections 2.1.1 and 2.1.2). We added a new module in COAWST–the *Rhizophora* root module–that provides the vertical profile of the projected area density of root systems from stem diameter and tree density in each model grid (Fig. 1; Section 2.1.3)."

**Reviewer [1.14]:**

line 110: "The Reynolds number …" – I am not sure if this was found in the publication (Shan et al) or in your work. Please clarify.

**Response [1.14]:**

It was found in previous studies. We have revised the said sentence as follows (L. 129–131 in the marked-up version):

> "The Reynolds number ($Re$) defined using the root diameter as length scale could be higher than the value ensuring fully turbulent structures of root-generated wakes ($Re$ > 120; Shan et al., 2019) even for weak currents (~1 cm s$^{-1}$) that could diminish the dependence of drag coefficient ($C_D$) on $Re$."

**Reviewer [1.15]:**

line 301: " We inputted …" Could this sentence be combined with the previous one? Now it seems like the same thing is said twice.

**Response [1.15]:**

We have combined the said sentence with the previous one as suggested (L. 325–327 in the marked-up version):

> "Among these, the proposed framework (Fig. 1) was used for the case $Rh$ model using the modeled $a_{root}$ (the *Rhizophora* root module provided the $a_{root}$ in the simulation) with input parameters of measured mean stem diameter ($D_{stem,ave}$) and tree density ($n_{tree}$)."

**Reviewer [1.16]:**

line 450: very long sentence. Please separate into two or three sentences.

**Response [1.16]:**

We have separated the sentence into three as suggested. Please see L. 484–489 in the marked-up version for the revision.

**Reviewer [1.17]:**

line 463: again very long, please separate.

**Response [1.17]:**

We have separated the sentence into two as suggested. Please see L. 497–500 in the marked-up version for the revision.

**Reviewer [1.18]:**

line 485: I think you can leave out the first sentence. As a reader I would not expect you to do more work so discussing what can be done in the future suffices here.

**Response [1.18]:**

We have removed the said sentence as suggested (L. 521–522 in the marked-up version).

**Reviewer [1.19]:**

line 504: I would acknowledge here studies that use dynamic vegetation models (which you cited in the foregoing line)

**Response [1.19]:**

We have added the reference as "(e.g., Xie et al., 2020)". Please see L. 542 in the marked-up version.

**Response to Reviewer #2**

**Reviewer [2.1]:**

**General Comments**

This manuscript presents a new approach to modeling the flow of water within Rhizophora mangroves. The key improvements to the COAWST vegetation package are: (1) allowing the vertical varying projected area density (frontal area per unit plan area), (2) using the root and stem length-scales in the turbulence dissipation terms, (3) implementing the Rhizophora module which can calculate projected area density from easily obtainable field measurements. These improvements allow the field to move beyond the conventional cylinder assumption, and are generally applicable to all hydrodynamically rough environments which aren't well described by cylinders.

I like the approach of this paper. The changes the authors have made have increased the clarity, and strength of this paper.

**Response [2.1]:**

We are grateful to the reviewers for taking the time to review our manuscript again and providing valuable comments.

**Reviewer [2.2]:**

The runs that are labeled as increased bed roughness (z0=0.02) should be considered with care because the z0 value used in those runs are an order of magnitude less than the authors estimate of the actual (without numerical limitations) increased bed roughness (z0=0.22). I think some text in the manuscript describing z0=0.2 as the maximum amount of bed roughness that logarithmic drag can represent in the model due to numeric limitation would be good. The authors thoroughly explain the numerical limitation in the supplemental information, I believe a few words in the manuscript would make it very clear to readers that z0=0.02 isn't the authors estimate of the enhanced z0 value. Alternatively, I believe that the requirement in COAWST that z0 < zbottom, is only true when using a logarithmic drag law. I have never attempted this, but I think it would be possible to use the equation (R8) relating the manning coefficient and C_{bed,mean} to arrive at a drag coefficient that can be input to COAWST using a quadratic drag law, getting around the z0 < zbottom limitation. Either additional text detailing the numerical limitations of and enhanced z0 value or a quadratic drag law approach would be sufficient in addressing this.

**Response [2.2]:**

We agree with the reviewer that some additional text describing the numerical limitation explains clearly the reason of choice for the value for $z_0$.

We have added the following sentence describing the numerical limitation in L. 342–344 (marked-up version) as suggested:

> "We note that the $z_0$ value equivalent to Manning's coefficient of 0.14 at 0.5 m water depth is $z_0 = 0.22$ m, but we were able to increase the value up to 0.02 m

due to the numerical limitation of the logarithmic velocity profile assumption implemented in the COAWST (Eq. (S13))."